# Trends in non-COVID-19 hospitalizations prior to and during the COVID-19 pandemic period, United States, 2017–2021

Kelsie Cassell [1] ✉, Casey M. Zipfel[2], Shweta Bansal[2] & Daniel M. Weinberger[1]

COVID-19 pandemic-related shifts in healthcare utilization, in combination with trends in non-COVID-19 disease transmission and non-pharmaceutical intervention use, had clear impacts on rates of hospitalization for infectious and chronic diseases. Using a U.S. national healthcare billing database, we estimated the monthly incidence rate ratio of hospitalizations between March 2020 and June 2021 according to 19 ICD-10 diagnostic chapters and 189 sub-chapters. The majority of primary diagnoses for hospitalization showed an immediate decline in incidence during March 2020. Hospitalizations for reproductive neoplasms, hypertension, and diabetes returned to pre-pandemic levels during late 2020 and early 2021, while others, like those for infectious respiratory disease, did not return to pre-pandemic levels during this period. Our assessment of subchapter-level primary hospitalization codes offers insight into trends among less frequent causes of hospitalization during the COVID-19 pandemic in the U.S.

The emergence of COVID-19 in the United States had immediate impacts on healthcare utilization and hospitalization rates due to non-SARS-CoV-2 related illness. Between March and April 2020, when a majority of US states enacted stay-at-home orders, non-COVID-19-related illnesses dropped approximately 20–30%[1,2]. These shifts in hospitalization rates could be attributed to a combination of factors influencing health and healthcare. During the early months of the pandemic, an estimated 41% of US adults avoided medical care due to COVID-19-related concerns[3]. At the same time, there were massive and long-lasting alterations to daily activities and contact patterns. Non-essential workers were required to work-from-home, businesses closed, students transitioned to remote learning, and outpatient healthcare settings favored telehealth visits over in-person visits when possible[4,5]. These sudden shifts, in addition to pathogen-pathogen interference, and non-pharmaceutical interventions (NPI) like face masks, had the potential to influence the transmission of non- SARS-CoV-2 pathogens and to both, directly and indirectly, affect the rates of hospitalization for a multitude of non-COVID-19 conditions[6,7].

The incidence of many infectious diseases decreased during the pandemic period (starting in March 2020) compared to previous years[8,9]. Influenza and respiratory syncytial virus (RSV) sharply declined during the winter of 2020/2021 and this was largely attributed to a decline in transmission of these viruses[10–12]. Conversely, certain conditions, like those related to mental health, were predicted to increase in incidence during pandemic shutdowns. Initial studies of self-reported depression and anxiety recorded during April 2020 supported this hypothesis, yet there has been mixed evidence on how this has translated to mental health-related hospitalizations[13–15]. Analyses of trends in non-COVID-19 disease during the pandemic period have highlighted the overall decrease in hospitalization rates during March and April 2020 and the unusual patterns in infectious respiratory disease; however, they are limited to specific subsets of health conditions, have short baseline comparison periods or are among specific sub-populations. A similar U.S.-based study was limited to Medicare beneficiaries and twenty non-COVID-19 conditions[16–21].

In this study, we leverage a large national healthcare billing database to estimate relative changes in the incidence of hospitalizations by diagnostic category as the COVID-19 pandemic emerged and progressed through the early-medium phase of the pandemic (first 16 months) and the first waves of SARS-COV-2 variants. Utilizing

[1]Department of Epidemiology of Microbial Diseases, Yale School of Public Health, New Haven, CT, USA. [2]Department of Biology, Georgetown University, Washington, DC, USA. ✉e-mail: kelsie.cassell@yale.edu

multiple years of baseline data, we provide robust estimates of trends in both common and relatively uncommon diagnoses among inpatients that are of interest to generalists and specialists alike. The results of this research provide a comprehensive view of disease-specific trends in relation to overall trends in hospitalization and can help to interpret reported trends in surveillance and hospitalization data.

## Results

The total number of unique hospitalized individuals ranged from 2.8 million to 3.1 million per year between 2017 and 2020. Between January 2017 and December 2019, the monthly number of hospitalizations (excluding repeat admissions occurring within a 30-day period and COVID-19 admissions) ranged from ~200,000 to ~450,000. In 2020, this number dropped from 455,000 hospitalizations in January to 263,000 to 224,000 encounters in April 2020, before rebounding moderately to 331,000 in June 2020. In total, 239,000 hospitalizations for COVID-19 (any diagnostic field) occurred during the study period (Supplementary Figure S3).

All ICD-10 chapters experienced relative increases in the incidence of hospitalizations during the pre-pandemic months of January and February 2020, with IRRs ranging from 1.08 to 1.21 in Jan 2020 (Fig. 1; Supplemental Table S3), likely reflecting database reporting changes at the start of the new calendar year. The exception was conditions of the perinatal period, which had an estimated IRR of 0.96 (95% CI: 0.87, 1.06) in Jan 2020. Beginning in March 2020, the relative incidence of hospitalization among all diagnostic chapters exhibited a moderate decrease compared to baseline. For most of chapters, the greatest declines in hospitalizations occurred during April 2020 (Supplementary Fig. S4).

This was especially true for diseases of the eye and diseases of the ear, while the smallest changes in hospitalizations were found among conditions classified within mental health, perinatal period, and pregnancy, childbirth, and puerperium chapters. The incidence of hospitalizations for respiratory disease was lower than expected for all months following April 2020, reaching the greatest relative decline in January 2021, when seasonal hospitalizations would typically be increasing (Jan 2021 IRR: 0.29; 95% CI: 0.28, 0.29).

Cluster analysis revealed three distinct groupings of ICD-10 subchapters (Supplemental Fig. S5). Subchapters grouped in cluster A included diagnoses for Pneumonia & influenza (P&I), Middle ear, Intestinal Infectious diseases, Conjunctivitis, Viral skin and mucous infections, Biomechanical lesions (M99), Acute lower respiratory infection (LRI), and Acute upper respiratory infection (URI) (Fig. 2; Supplemental Fig. S6). These diagnoses shared a sharp relative decrease in hospitalizations in April 2020 that continued to have below-expected rates throughout the remainder of 2020. The average IRR of this cluster was 0.46 (range: 0.11, 1.09) between March 2020 and June 2021. The incidence of P&I hospitalizations in January and February 2021 was 89% lower (95% CI: 88.7%, 90.4%) than what would be expected given predicted trends in hospitalizations from previous years and seasonal trends (Supplemental Table S4). There were only 9,427 unique hospitalizations for P&I between Oct 2020 and Mar 2021 compared to 61,085 during the same months of 2019. The timing and intensity of the decline in hospitalizations varied by subchapter within this cluster, with most subchapters experiencing the greatest decline in April 2020 or January 2021. Subchapters Intestinal infectious diseases, Acute URI,

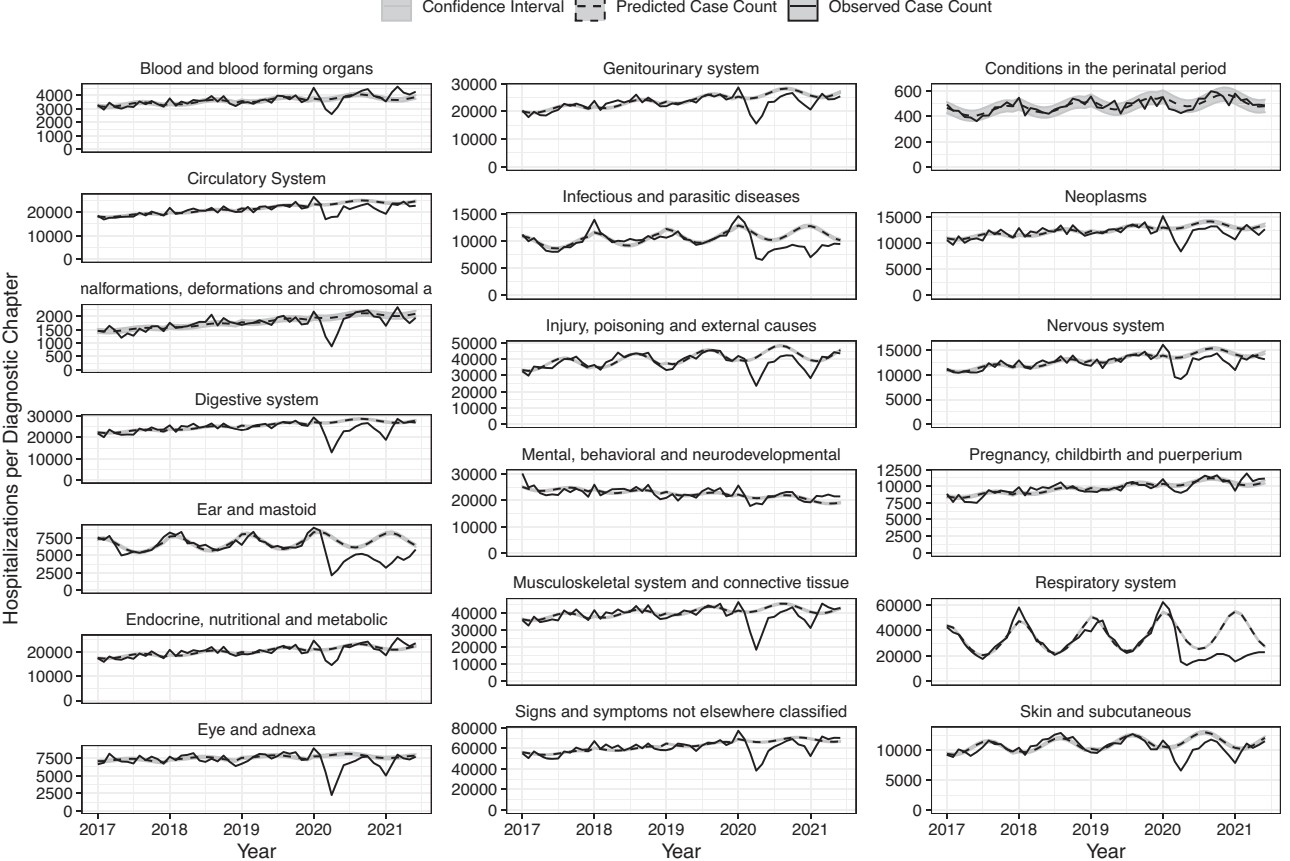

**Fig. 1 | Time series of observed hospitalizations per diagnostic chapter by month and year (solid line) with model predicted case counts (dashed line) and prediction intervals (grey).** Note: Data are presented as the observed case count per month (black), with regression model predicted case counts (dashed black) surrounded by a gray prediction interval estimated through two-stage simulation using Monte Carlo resampling that accounted for parameter uncertainty and observation uncertainty.

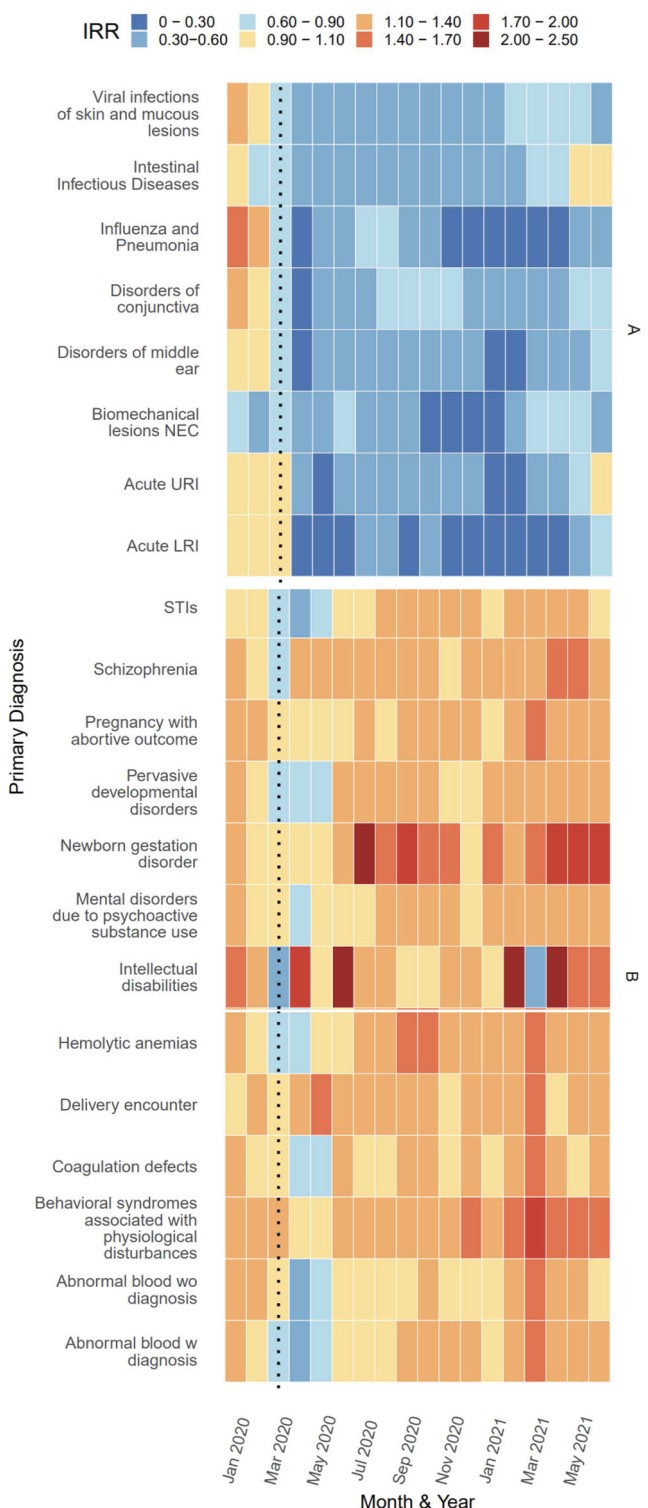

**Fig. 2 | Estimated Incidence Rate Ratio (IRR) per subchapter and month in Clusters A and B, Jan 2020 to Jun 2021.** Note: Estimates of IRR made by dividing observed by predicted cases per diagnosis per month. Monthly predictions made using trends in previous years data after adjusting for seasonality and time trends with an offset for estimated active population. Prediction intervals for estimates show in Supplementary Figure S6.

and Acute LRI approached pre-pandemic incidence rates (i.e. IRR approaching 1.0) in May and June of 2021.

Diagnoses within cluster B included: Sexually Transmitted Infections (STIs), intellectual disabilities, psychological disturbances,

and coagulation defects, in addition to ICD-10 codes related to pregnancy such as delivery encounter, pregnancy with an abortive outcome, and newborn gestation disorder. In contrast to cluster A, cluster B had little to no decrease in hospitalizations during March, April, or May of 2020. The average IRR in this cluster remained at or below expected levels from March 2020−May 2020 and exceeded expected levels for the remainder of the study period. The average IRR of this cluster between March 2020 and June 2021 was 1.20 (range: 0.46, 2.20). Certain diagnoses within clusters A and B, like those for intellectual disabilities, physiological disturbances, and biomechanical lesions, show significant deviations from the expected hospitalization rate; however, the frequency of events was low.

Cluster C included all remaining diagnoses assessed. Diagnoses for hypertension, malnutrition, diseases of the appendix, diabetes, and neoplasms of multiple systems (e.g. respiratory, CNS, reproductive organs, soft tissue, breast, digestive organs, etc.) decreased in hospitalizations during March, April, and May of 2020, well as January of 2021, but otherwise hovered around pre-pandemic estimates of IRR (Figs. 3, 4). Diseases of the lens, which experienced the largest drop in hospitalizations during April 2020 (IRR: 0.12; 95% PI: 0.12, 0.13) before returning to its pre-pandemic range by June 2020. In contrast, diagnosis for pulmonary heart disease, ischemic heart diseases, injuries of multiple areas of the body (thorax, shoulder, neck, knee, hip, head, hand, foot), chronic lower respiratory disease, chronic rheumatic fever, and neoplasms of the skin exhibited a relative decrease in hospitalizations during March, April, and May 2020 and continued to exhibit lower-than-expected hospitalizations during the remainder of 2020 and early 2021 with occasional months near baseline (Fig. 3, 4).

Sensitivity analysis that excluded subsets of payers showed similar estimates to those calculated using data from all payers (Supplemental Table S5–S6). The largest differences between the original results and those of the sensitivity analyses were among rare disorders (e.g. Intellectual disabilities, Newborn gestation disorder, Physiological disturbances) during the early months of 2021.

## Discussion

The COVID-19 pandemic has profoundly disrupted healthcare and disease dynamics in the United States. The wealth of data provided by this healthcare insurance billing database allowed for a large-scale examination of these disruptions among a vast range of disease categories. The majority of categories experienced a decline in relative incidence of hospitalizations during April and May of 2020, while the timing at which incidence returned to pre-pandemic levels varied by diagnostic chapter, and some chapters did not reach pre-pandemic levels of hospitalization incidence by the end of the study period.

Hospitalizations with a primary diagnostic code for infectious respiratory disease (e.g. influenza, acute lower respiratory infections) and otitis media (e.g. "middle ear diagnosis" subchapter) clustered in our analysis given their shared dramatic decrease in IRR during April and May 2020 and their uncharacteristically low relative incidence during fall/winter 2020/2021. These results substantiate prior studies of influenza, RSV and other upper and lower respiratory infections during the winter 2020/2021 season[10,22]. While the decline in influenza could be partially attributed to a decrease in the number of specimens sent for respiratory panel testing, as evidenced by an 61% decrease in US specimens sent for testing in 2020, studies have shown that the positivity rate among specimens sent had also declined, approximately 98%, during winter 2020/2021[11].

In contrast to the subchapters with large declines in IRR, the cluster of subchapters with increased rates of hospitalization (cluster B), included diagnoses for behavioral syndromes associated with physiological disturbances, hemolytic anemias, and intellectual disabilities. Interestingly, these illnesses did not appear to have a clear shared mechanism to explain the pandemic-period trends. Potential drivers of the sporadic increases among any disease incidences during

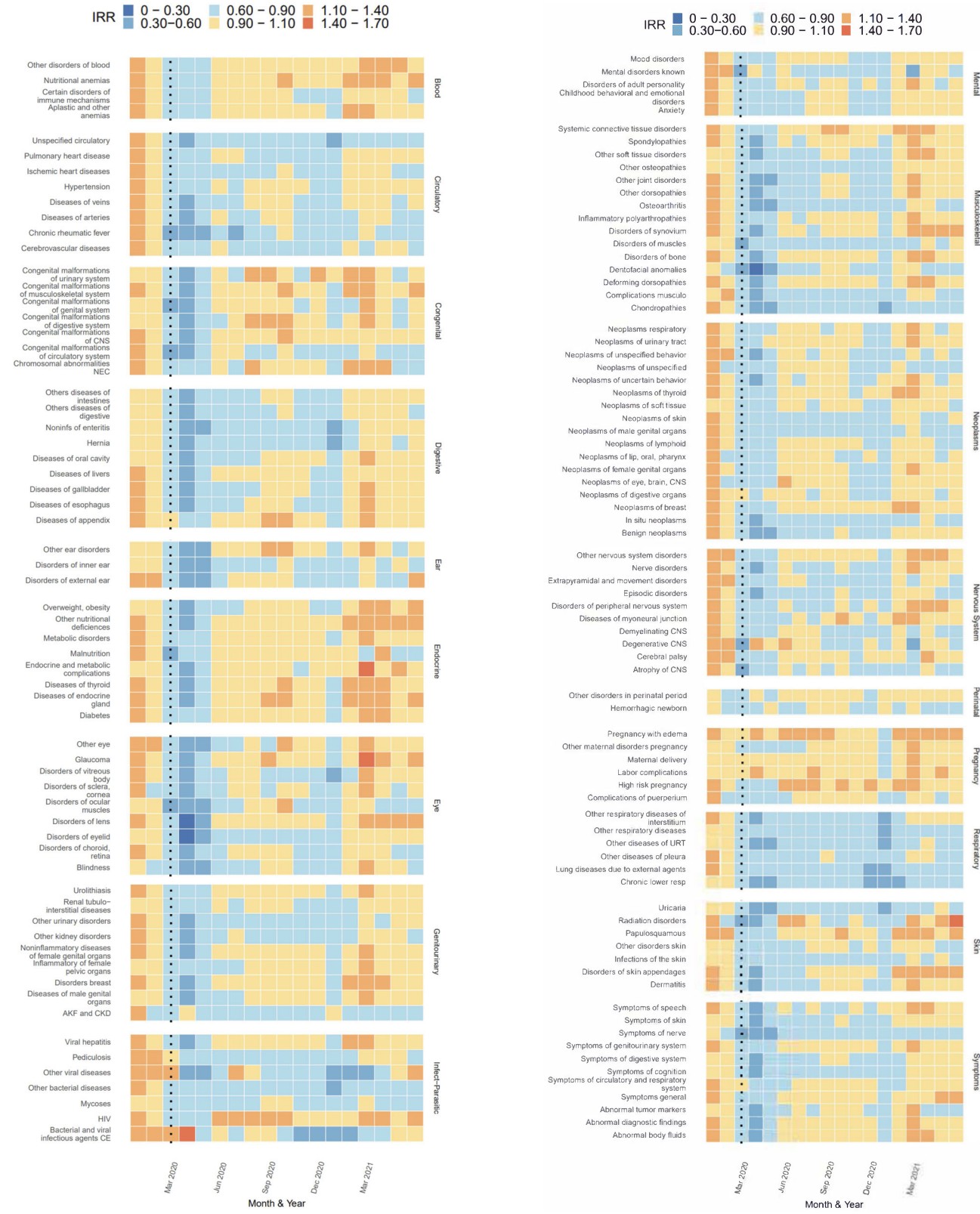

**Fig. 3 | Estimated Incidence Rate Ratio (IRR) per subchapter and month from cluster C grouped according to diagnostic chapters (Blood, Congenital, Digestive, Ear, Endocrine, Eye, Genitourinary, and Infectious-Parasitic), Jan 2020 to Jun 2021.** Note: Estimates of IRR made by dividing observed by predicted cases per diagnosis per month. Monthly predictions made using trends in previous years data after adjusting for seasonality and time trends with an offset for estimated active population.

**Fig. 4 | Estimated Incidence Rate Ratio (IRR) per subchapter and month from cluster C grouped according to diagnostic chapters (Mental, Musculoskeletal, Neoplasms, Nervous System, Perinatal, Pregnancy, Respiratory, Skin, and Symptoms), Jan 2020 to Jun 2021.** Note: Estimates of IRR made by dividing observed by predicted cases per diagnosis per month. Monthly predictions made using trends in previous years data after adjusting for seasonality and time trends with an offset for estimated active population.

the pandemic period include: (1) changes in the underlying population covered by the insurance payers included in our analysis or changes in which hospitals (or types of hospitals that) are within the insurance service areas, (2) diagnoses that are a direct result of a prior SARS-CoV-2 infection, and (3) mechanisms that cannot be elucidated using this dataset such as diagnostic screening, test ordering, and the prioritization of hospitalization vs outpatient care settings.

Similar to other studies of non-COVID-19 trends, we find an immediate relative decline in hospitalizations for neoplasms that rebounded later in 2020, followed by a shorter period of a relative decline in the winter of 2020/2021[18]. The largest decrease was among benign neoplasms in April 2020 (−77%, PI: −76%, −78%). Neoplasms of the skin, thyroid, and male genital tract also had a decrease in hospitalizations of 30% or more in April 2020. The most likely explanations for the fluctuations in neoplasm diagnoses are disruptions to cancer screenings and subsequent diagnoses resulting from healthcare access limitations during the early pandemic period followed by an influx of patients in need of delayed cancer screenings and referrals[23–26].

Primary hospitalizations for injury related hospitalizations remained below expected rates in 2020, which is in apparent contrast to results from a similar study conducted in Denmark wherein hospitalizations for injuries appeared to return to expected incidence levels[18]. As our analysis relies on inpatient data, we are likely only seeing a small portion (and severe enough to require an inpatient hospitalization) of the total injuries that were presenting during this time. Because countries had differing approaches to pandemic-related shutdowns and triaging of patients, it's possible that the timing of rebounds for non-infectious disease hospitalization will not perfectly align from country to country.

The ICD-10 chapter for Pregnancy, Childbirth, and Puerperium only experienced an estimated decrease in relative incidence of 11% (95% PI: 9%,13%) in April of 2020. As expected, admissions for maternal delivery, did not fluctuate during the pandemic period; however, birth outcomes could be influenced by the level of prenatal care and pregnancy-related stress as the pandemic continued into late 2020 and early 2021. Throughout the study period, interesting signals were noted among hospitalizations for newborn gestation disorder and pregnancy with an abortive outcome. Incidence was aberrant and did not appear to clearly alter in relation to COVID-19 pandemic changes in health care. The increase in incidence during 2020 and 2021 for these diagnostic codes could be a product of an otherwise increasing trend in data reporting to the database or a true reflection of pregnancy/newborn outcomes related to the COVID-19 pandemic.

Cluster C, as a whole, could be characterized as showing declines in relative incidence in correspondence with waves of COVID-19 occurring in the US, primarily March and April of 2020 and January 2021. This supports the hypothesis that these hospitalizations are primarily those that could be delayed during peak COVID-19 transmission cycles. A Spanish study that included "factors influencing contact with health services" as a diagnostic chapter of interest, showed a relative increase during Sept and Oct 2020 which could reflect delayed care-seeking behavior[20].

Due to the passive nature of data reporting to the healthcare encounter billing clearinghouse and potential differential reporting by insurance payer, different payers may contribute to relative increases or decreases in disease rates in a manner that might not represent generalizable trends across all areas of the US during the study period. Alternative explanations for the relative increases in disease diagnoses include payer catchment area redefinitions (and sudden inclusions of specialty facilities), reimbursement policies, and primary vs secondary diagnostic field coding practices. However, sensitivity analysis, where 20% of the top 100 payers were excluded from analysis, revealed similar results, suggesting that large payers were not erroneously driving specific diagnostic category results. For most diagnostic chapters and subchapters, there was an increase in hospitalizations in

Jan and Feb 2020 which indicated a fluctuation in the billing data reported to the database used for this analysis. This complicates the interpretation of elevated incidence rates among certain diagnostic subchapters during the pandemic period as it may be a reflection of the overall influx of data to this clearinghouse that began at the start of the 2020 calendar year.

Our analysis was limited to primary diagnostic codes and it is possible that IRR trends may vary if the full breadth of secondary codes were included for analysis. This study was also limited to inpatient data; therefore, we are unable to discern if relative increases or decreases in hospitalization may be linked to a shift in treatment location (i.e. cases being treated in an outpatient setting that would otherwise be hospitalized or vice versa). Similarly, we are unable to make inferences on the prevalence or incidence of disease in the general population, since the threshold for hospitalization may have changed during the pandemic period and hospitalization rates of disease rarely match incidence rates of disease in the general population[27]. This phenomenon has been noted among cases of appendicitis such that more cases were treated with non-operative management prior to delayed operative treatment during the early pandemic period[28,29]. Unfortunately, the database used for analysis did not include gender, or race/ethnicity information, thereby limiting generalizability interpretations. Because the database was not updated after September 2021, analyses of the latest waves of the pandemic were not possible and later trends may vary from early trends in non-COVID-19 hospitalizations.

This study is strengthened by the large population size (including U.S. patients on public and private insurance) and the ability to investigate rare disorders that may otherwise not be captured in an analysis of disease trends using localized data sources. Our ability to access multiple years of data prior to the pandemic also strengthens our confidence in estimates of relative incidence during the pandemic period. Examination of billing data allowed this study to overcome limitations in disease reporting due to COVID-19 related disruptions, as has been demonstrated for the Nationally Notifiable Disease Surveillance System (NNDSS) data[8].

There were substantial changes in hospitalization patterns across many different disease categories during the early part of the COVID-19 pandemic, some of which persisted for at least a year. The broad trends identified here point to a number of hypotheses about the mechanisms driving these trends. Declines in infectious respiratory disease may likely be a result of NPI use, social distancing, and changes to daily life that altered contact patterns. Meanwhile, this research has highlighted potential areas for further investigation by experts in the field, such as trends in hospitalizations for intellectual disabilities, newborn gestation disorders, and mental health disorders. We hope that specialists and generalist physicians alike, as well as epidemiologists, can find utility in this study when analyzing their own data, or dissecting trends in rare diseases.

## Methods
### COVID-19 Research Database
Data were obtained from the *COVID-19 Research Database* (C19RDB), which is a pro-bono public-private consortium that collects and aggregates de-identified healthcare data, such as healthcare claims data. For this study, we used a health insurance claims (i.e. billing) database (specifically, the OfficeAlly clearinghouse), which contains data on healthcare interactions of patients for all causes (not just COVID-19) from both outpatient and inpatient interactions. We focus on only inpatient interactions within our study. The database included identifiers for healthcare setting (inpatient, outpatient), primary diagnostic code, secondary diagnostic codes, billing site location (state), date of service, Current Procedural Terminology codes, age and gender, among other billing details. Reporting to this database was continually updated until September 2021, leading to variation in total

claims per month as additional payers were incorporated over time (Supplemental Figure S1). This inpatient data specifically captures billing data at the level of each individual hospitalization per anonymized subject ID number. Information on the total population within each insurance payer (i.e. payees) was not available, limiting our ability to accurately assess a population denominator from which hospitalizations arose[30]. Insurance payers covered patients from all fifty states and Puerto Rico, but some payers covered patients in multiple states and regions without specifying which state a patient resided in, thus information is not presented by state or region. This research has been approved by the Georgetown University Institutional Review Board for Human Subjects Research.

## Data cleaning

We limited the data to inpatient records due to large and unexplained fluctuations in the volume of outpatient data entering the database in early 2019. Only hospitalizations with a corresponding procedure code were included for analysis. Entries were deduplicated for each anonymized ID to retain only the first hospitalization record within a 30-day rolling period for each patient. The month and year of diagnosis used for analysis was the "service from date" when available, otherwise "service to date" or "admission date" or "billing statement from" date was used (in order of first availability). "Service from date" was present in the majority (>90%) of records and was the primary source for date information. Hospitalizations for COVID-19 were excluded from all analyses if a code for COVID-19 (ICD-10 "U07.1−COVID-19" or "J12.82−pneumonia due to SARS-associated coronavirus" or "M35.81−COVID-19 related multisystem inflammatory syndrome" or "B97.2*−coronavirus as the cause of disease elsewhere classified") was present, in order to properly assess trends in non-COVID-19 illnesses[31]. The final study period was January 1, 2017 to June 31, 2021 (Supplementary Fig. S2).

The outcome of interest was the primary (i.e. first) diagnostic field recorded for each visit as categorized by the International Classification of Diseases-Clinical Modification (ICD-10-CM) codes. We included only the primary diagnostic code for each hospitalization in order to limit the influence of diagnoses not directly related to admission[32]. Primary diagnostic codes for all patients were grouped by ICD-10 chapter (2016 edition of ICD-10-CM), subchapter, month, and year. These chapters are primarily organized according to body system affected, totaling 22 chapters, of which diagnostic codes falling into chapters 1 through 19 were assessed (excluding chapters for external morbidity, special purpose codes, and health services contact codes) (Supplementary Table S1). Subchapter groupings were created according to categories defined by the World Health Organization (WHO) and were primarily organized by type of infectious agent identified, type of illness, or part of body affected in addition to codes for "unspecified" or "unknown" illness[33]. The 19 chapters could be subdivided into 236 subchapters, but subchapters with fewer than 5,000 diagnoses total between 2017 and 2019 were excluded from all analyses (47 exclusions) (Supplementary Table S2).

## Analysis

The aim of this analysis was to provide a monthly estimate of the incidence rate ratio (IRR), defined as the observed number of monthly hospitalizations divided by the expected monthly hospitalizations, for each diagnostic chapter and subchapter of interest. To estimate the number of hospitalizations per month had the pandemic not occurred, Poisson regression models were fit to the monthly count of diagnostic codes within each ICD-10 chapter (or subchapter) from the pre-pandemic period (January 1, 2017−December 31, 2019). Time series for each chapter and subchapter were analyzed independently. All models were adjusted for time trends (linear trend) and yearly seasonality (using sine and cosine harmonic terms). While we did not have a true population denominator to use as an offset term in the model, due to a lack of information on payer population or hospital catchment areas, we estimated the size of the active patient population as the number of unique individuals with a recorded hospitalization per year for any cause. For the hospitalizations occurring between 2017 and 2019, the offset was the total number of unique individuals present in the same calendar year. Because there was a generally increasing trend in the number of individuals included in this database, the offset for 2020 and 2021 was estimated by fitting linear and seasonal trends to the monthly counts of unique individuals in the study prior to March 2020 and extrapolating these trends to the period from March 2020-June 2021.

Prediction intervals for the final IRR estimates were calculated for each time point through a two-stage simulation using Monte Carlo resampling that accounted for parameter uncertainty and observation uncertainty[34,35]. In the first stage, we resample from a multivariate normal distribution, using the mean and variance-covariance matrix of the regression coefficients. These samples are combined with the covariates to get samples of the regression mean. We then take samples of these samples, where the regression means are drawn from a Poisson distribution. This adds observation uncertainty. The result of this resampling scheme is a 95% prediction interval. Sensitivity analysis were conducted wherein a random selection of 20 of the 100 largest (by unique patients) payers were excluded from analysis. This process was repeated 1000 times and the median IRRs (95% sample interval) for all ICD-10 chapters and subchapters of interest.

Due to the numerous subchapters included in this analysis ($N = 180$), we implemented a hierarchical clustering algorithm (applying Ward's minimum variance method) to systematically group diagnoses that shared temporal trends in IRRs during the period between Jan 2020 and Jun 2021.

Data queries and analyses were run in SQL and R version 4.0.2 via Amazon Workspace using the packages cluster-2.1.0, data.table-1.13.0, DBI-1.1.0, dplyr-1.0.2, ggplot2-3.3.2, glue, gridExtra-2.3, lubridate-1.7.9, patchwork-1.0.1, readr-1.3.1, stringr-1.4.0.

## Reporting summary

Further information on research design is available in the Nature Research Reporting Summary linked to this article.

# Data availability

The data that support the findings of this study are subject to controlled access due to the inclusion of protected health information in the dataset and restrictions on use set by the COVID-19 Research Database consortium. Access to data is permitted upon application for use to the database and approval, via contact@covid19researchdatabase.org. Responses to data requests are expected to take 3 months and access is limited to academic, medical, and scientific research of COVID-19 only.

# Code availability

COVID-19 Research Database has prohibited the public sharing of code that was used to clean or generate the analysis used in this study for confidentiality reasons. COVID-19 Research Database states: "[The] database is housed within a secure Amazon Workspace (AWS) Virtual Environment. All analyses are conducted within the AWS environment. To ensure the security and privacy of the data, no code or data can be brought into the environment and no code can leave the environment."

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

## Acknowledgements

The data, technology, and services used in the generation of these research findings were generously supplied pro bono by the COVID-19 Research Database partners, who are acknowledged at https://covid19researchdatabase.org/. We would also like to thank Dominic Copper-Wooten for his contributions regarding SQL querying. KC would like to acknowledge funding by the National Institute of Allergy and Infectious Diseases of the National Institutes of Health under award numbers # 1F31AI161971-01A1. SB would like to acknowledge the Merck Investigator Studies Program, Award # 60274, and National Institute of General Medical Sciences of the National Institutes of Health under award number R01GM123007.

## Author contributions

Concept and design: S.B., D.M.W., K.C. Acquisition, analysis, and data interpretation: S.B., K.C., D.M.W., C.Z. Statistical analysis: K.C., D.M.W. Drafting of the manuscript: K.C., D.M.W., S.B. Manuscript revision and final approval: K.C., C.Z., D.M.W., S.B.

## Competing interests

D.M.W. has received consulting fees for work unrelated to this manuscript from Pfizer, Merck, GSK, Affinivax, and Matrivax, and is Principal Investigator on research grants from Pfizer and Merck to Yale University for work unrelated to this manuscript. S.B. is Principal Investigator on a research grant from Merck to Georgetown University. The remaining authors declare no competing interests.
