## [Peer Review File · Nature Communications]

Trends in non-COVID-19 hospitalizations prior to and during the COVID-19 pandemic period, United States, 2017 – 2021REVIEWER COMMENTS

Reviewer #1 (Remarks to the Author):

It has been previously reported that dramatic changes in healthcare utilization have occurred due to the COVID-19 pandemic, including changes in hospitalizations for non-COVID related reasons. This is due to many factors, including pandemic shutdowns, social distancing and face masks, patients avoiding hospitals for screening tests, delays in non-essential procedures, etc. However, the authors state that prior work in this area has focused on individual conditions or smaller populations. This study takes a systematic look at this issue by estimating changes in hospitalizations across all ICD-10 diagnoses for a large U.S. population of 13.5M patients with 19.2M hospitalizations. Diagnoses were grouped into four clusters, with results similar to prior studies. For example, non-COVID infectious diseases in cluster A immediately fell in March 2020 and remained well below normal for the rest of the study period, ending in June 2021.

Understanding how the pandemic affected non-COVID-19 hospitalizations is essential for many applications. There actually have been other studies like this that looked systematically across diseases for large populations. For example:

In Denmark:

Bodilsen J, Nielsen PB, Søgaard M, Dalager-Pedersen M, Speiser LOZ, Yndigeegn T, Nielsen H, Larsen TB, Skjøth F. Hospital admission and mortality rates for non-covid diseases in Denmark during covid-19 pandemic: nationwide population based cohort study. *BMJ*. 2021 May 24;373:n1135. doi: 10.1136/bmj.n1135. PMID: 34035000; PMCID: PMC8142604.

In Spain:

Pifarré I Arolas H, Vidal-Alaball J, Gil J, López F, Nicodemo C, Saez M. Missing Diagnoses during the COVID-19 Pandemic: A Year in Review. *Int J Environ Res Public Health*. 2021 May 17;18(10):5335. doi: 10.3390/ijerph18105335. PMID: 34067807; PMCID: PMC8156815.

In the Philippines:

Uy J, Siy Van VT, Ulep VG, Bayani DB, Walker D. The Impact of COVID-19 on Hospital Admissions for Twelve High-Burden Diseases and Five Common Procedures in the Philippines: A National Health Insurance Database Study 2019-2020. *Lancet Reg Health West Pac*. 2022 Jan;18:100310. doi: 10.1016/j.lanwpc.2021.100310. Epub 2021 Nov 3. PMID: 34751261; PMCID: PMC8565915.

In the US (although not broken out by disease category, this uses a large 8M patient U.S. Medicaid database):

Dang A, Thakker R, Li S, Hommel E, Mehta HB, Goodwin JS. Hospitalizations and Mortality From Non-SARS-CoV-2 Causes Among Medicare Beneficiaries at US Hospitals During the SARS-CoV-2 Pandemic. *JAMA Netw Open*. 2022 Mar 1;5(3):e221754. doi: 10.1001/jamanetworkopen.2022.1754. PMID: 35262712; PMCID: PMC8908076.

Thus, while the systematic look across all diseases is not novel, I do think that the large size of this study adds new accuracy and details that haven't been seen before.

Major Concerns

- 1) A key difference between this study and others is the use of the large COVID-19 Research Database. However, the manuscript says very little about this. Much more detail is needed.
 - 1.1) I am aware of the COVID-19 Research Database, but I know very little about it. My understanding is that it is a collection of databases contributed by different organizations to support COVID-19 research. I think the authors selected a single claims database, Change Healthcare, which combines data from multiple insurance companies/payers. However, the authors do not actually name the database. Is there a reason for this? Why did they select this database?
 - 1.2) I know a couple of groups that have been trying to use the Change Healthcare data in the COVID-19 Research Database, but they have run into many problems. Can the authors either reference another study that has used this data or provide some information about how they did it

(e.g., the process for getting access to the data, what software programs they used for the analysis, etc.). This could be very helpful to others.

1.3) How many different payers are represented in this database? The authors wrote "additional payers were incorporated over time". Did payers ever leave the database?

1.4) It would be helpful to at least have a summary figure showing (a) the total number of hospitalizations in this database per month, (b) the number of COVID-19 and non-COVID-19 hospitalizations per month, and (c) the number of payers included in the database per month. Seeing the number of payers per month could provide insight into the strange spikes in Jan 2020 and Jan 2021.

1.5) What are the age/sex/race/ethnicity distributions in the data?

1.6) The pandemic waves hit different parts of the U.S. at different times, and healthcare utilization was probably affected in different ways by location. For example, masks were used more in some places than others, pandemic shutdowns varied by location, etc. The authors note that their data covered all fifty states and Puerto Rico. However, the distribution is not reported. Because the insurance payers changed over time, the geographic distribution might also have changed. Ideally, the authors could provide some information, at least by region of the country, on coverage by month. If not, then they should comment about this.

1.7) The Data Availability statement indicates the data can be obtained from the COVID-19 Research Database consortium. However, I'm not certain I know which subset of that database the authors used for this study. I'm guessing Change Healthcare. Can the authors share the code they used? I've heard that it is difficult to write the SQL queries for Change Healthcare. It would be helpful to others to see how they did this. The authors note that they are "in the process of gaining approval...to make the relevant code public."

2) It would be good to have a figure like Figure 1, but one graph for each cluster (combining all hospitalizations within the cluster). It is difficult to see the overall trends in Figure 2, where it is broken out by subchapter.

3) On page 4, line 155, the authors write that they used a "two-stage simulation using Monte Carlo resampling". They give a couple references, but a brief summary of what this is would be nice.

Minor Comments

4) Figure 2A is for clusters A and B, right? If so, then this should be noted in the caption.

5) In Figure S1, the de-duplication step increased the number of records from 33.6M to 34.1M. Is this a typo? How did removing duplicates increase the number?

6) Is the caption for Figure S3 correct?

Reviewer #2 (Remarks to the Author):

General comments:

This is an interesting epidemiological study of the estimated effects of the COVID-19 pandemic on non-COVID-19 hospitalization trends between the beginning of the pandemic and June 2021. The use of a large database with 3 years of hospitalization data prior to the start of the pandemic combined with a thorough analysis results in a convincing depiction of these trends. The most significant limitation is the lack of data beyond June 2021 – which means that there is no information on trends during the Delta and Omicron waves, which were both extremely significant, but also occurred in a population who had been living with the pandemic for over a year already (see attached image file). In addition, vaccination began in December 2020, and as a result most of this study's period of interest is in the pre-vaccination period, which would obviously have a huge impact on population-level behavior.

The result of this limitation may be that this manuscript does not hold the practical utility the authors suggest, and the focus should be more on the interesting epidemiological findings of the detailed effects of a large pandemic on non-COVID hospital services.

Furthermore, due to the methodology and types of data used, it is difficult to ascertain causality: a whole range of things changed with the onset of the pandemic and teasing apart which factor influence the drop is not possible from this type of analysis. As such, once again, the language could be focused more on describing the epidemiological findings than on possible causative mechanisms (specific points in the comments below).

Detailed comments:

Introduction:

Lines 72-75: The two articles referenced here were published at a much earlier stage in the pandemic. Do the results of the current manuscript (that show trends up until June 2021 – before Delta and Omicron and in a predominantly pre-vaccination period) really aid us in hospital preparation and resource allocation? If so, how?

Lines 79-81: The analysis results are limited in their ability to inform such specific hypotheses – in order to inform the hypothesis for example that NPIs influenced hospitalizations numerous confounders would have to be dealt with, and this is out of the scope of the current article. Consider removing this sentence.

Methods:

Lines 114-118: Does this include primary as well as secondary diagnostic codes? The number of hospitalizations excluded (66,099) seems a little small on the basis that we are drawing from a population significantly larger than 13.5 million individuals who were hospitalized in this dataset. From the CDC website it appears as if about nearly 2.5 million hospitalizations occurred in the US by June 2021 (<https://www.cdc.gov/coronavirus/2019-ncov/covid-data/covidview/index.html>). Maybe some of the hospitalizations removed because they occurred within a 30-day window of prior hospitalization were COVID-19 related?

Line 133: Please note that in Supplementary Table S2 it is written that subchapters with fewer than 500 diagnoses (rather than 5000) were excluded.

Overall, the development of the Poisson models to predict the expected rates based on the previous three years seem to have been performed rigorously and provides a good estimated baseline for expected rates.

Results:

Lines 171-177: The phrasing here is confusing. In general the monthly unique hospitalizations including the relevant trends should be described from Jan 2017 until Feb 2020 (including an additional mention of a large increase in Jan and Feb 2020 if indeed there was a big jump to 455,000 from less than 400,000 in any month prior to 2020) and then a description of the general trends from March 2020 until June 2021. This information is not presented in a figure or table (at least no figure or table is referenced) and as such needs to be clearly stated here in full. It might be also useful to note here the number of excluded COVID-19 related hospitalizations during the pandemic period.

Lines 181-183: It is confusing to report relative increases across all groups and then present an IRR of 0.96 which would represent a decrease. Consider specifically commenting on Conditions in the preinatal period as the only chapter that didn't show an increase.

Lines 184-185: Likewise with March 2020, the perinatal period chapter is the same in March as February and as such this sentence is confusing.

Lines 196-243: The description of the clusters is a bit clunky and could be streamlined. Maybe consider leading with the types of conditions in each cluster followed by the trends of that cluster of the pandemic. For example: Cluster A consists of the subchapters [list relevant subchapters]. These subchapters demonstrated a sharp relative decrease in hospitalizations early in the pandemic with rates below expected until [the end of 2020? June 2021?]. Then provide some specific details of interest such as rates of P&I. It may help with the flow and clarity of the clusters.

Lines 218-225: Specifically with cluster B, the phrasing is confusing – consider “the average IRR in this cluster remained at or below expected levels from March 2020 – May 2020 and exceeded expected levels for the remainder of the study period.

Line 242-243: Please update to ‘between March 2020 and June 2021’. Current phrasing ‘between March and June 2021’ is confusing

Discussion:

Lines 276-318: Generally this section needs to be cut down. A significant proportion of the text is spent trying to explain what seem like possible artefacts in the data or things that the data cannot explain such as the trends in hospitalizations where the primary diagnosis is intellectual disabilities. It might be more effective to build a paragraph around ‘signals’ generated that could be leads for further research (is there a causal relationship between SARS-CoV-2 pandemic and hemolytic anemias/intellectual disabilities/adverse newborn outcomes). In addition, as a way of providing some validation to the results, it would be interesting to show that the number of childbirth coded diagnoses (i.e. sub-chapter) didn’t drop in the early pandemic (it should be unaffected overall as the authors mention) and that the drop in the pregnancy, childbirth and puerperium diagnoses occurred as a result of codes from the pregnancy and puerperium related codes.

Conclusion:

Lines 380-387: The conclusion here focusses a lot on the aspects that the analysis cannot actually answer – why the trends are what they are. An alternative focus for the conclusion could be that for the first time, by leveraging a large billing database in the US, a comprehensive high-level analysis of hospitalization by diagnosis has identified broad trends over the pandemic period. This is useful because...[insert reasons].

Lines 387-388: It is unclear if the last sentence adds value to the manuscript and if so, how this will help minimize disruptions during future major health events. Recommend removing and leaving the reader to decide if the findings are of interest/use.

Miscellaneous:

Supplementary figure S3 – The legend seems to be unedited: “shows the clusters of the subchapters and where I made the cut-offs – can update this to have clear demarcations of the cut-offs” – please correct.

Reviewer #3 (Remarks to the Author):

Thank you for giving me the opportunity to review this manuscript examining changing rates of hospitalisation for a range of conditions in the USA between 2017 and 2021.

The article uses an administrative healthcare claims dataset which represents a non-random sample of data provided by a range of primary and secondary healthcare institutions. The authors use these data to estimate monthly incidence rate ratios for hospitalisation for a panel of conditions derived from the ICD-10 codes of the primary diagnosis of an admission. The authors then use hierarchical clustering to identify four clusters of conditions whose changing incidence rate ratios are similar to one another.

This is an interesting study that leverages a large dataset to look at month to month changes in incidence, in addition to looking at a large number of conditions. The differences in trajectories for the clusters identified are interesting and may provide an important insight into the ways health systems provided care to patients with different conditions.

I do have several concerns regarding the dataset, methods used and the way in which findings are interpreted.

1. The underlying dataset used.

The authors acknowledge that the C19RDB dataset is imperfect, relying on the voluntary contribution of claims data from providers over time and resulting in sometimes large fluctuations in the amount of data received. It is not clear the extent to which these fluctuations are random with respect to the underlying features of the data received, whether with respect to centres providing data and the characteristics of their patients. This also leads to a lack of a population denominator for their models, which the authors acknowledge. It would be useful to understand the sensitivity of their models to changes in the assumptions of the active patient population (lines 146-149).

2. Clustering methodology.

The article relies heavily on the identification of clusters of conditions and the discussion consists mostly of discussion of these clusters and their collective temporal trends. As such, I would hope to have far more detailed explanation of the clustering process used, considerations made on methods used and how data were prepared prior to clustering. Was each month treated as an independent data point, or were efforts made to otherwise characterise the time series in a higher order manner by relating each data point to its neighbours or similar? Why was hierarchical clustering chosen over other methods? Not to say it's a bad choice per se, but the authors also use K-means clustering alongside and then choose to adopt the clustering derived from a hierarchical method.

I'm not sure about the appropriateness of using K-means clustering with an elbow plot to justify the choice of cluster number for a hierarchical clustering output. Both are different methods and may have different optimal partitions. The dendrogram should be enough to determine which of many possible configurations is notionally optimal for the purpose of the analysis.

There is no clearly optimal clustering configuration to the data, as there rarely are outside of purely artificial datasets. That said, looking at the PCA plot, it seems like the 4 cluster configuration is perhaps inappropriately splitting the central mass of the point cloud. This is perhaps backed up by Figure S3 in which either 3 or 7 clusters looks to be the best partition of the heat map. The elbow plot is a pretty gentle transition, with barely an elbow at all, so I'm not sure it strongly supports a choice of 4 clusters. 3 clusters is perhaps the right split to avoid overselling difference observed between trajectories.

Given the central role of the clustering process to the paper, I would expect more of the clustering figures to feature in the main dataset, e.g. Figure S3.

3. Interpretation of findings.

In the discussion, the authors explore the changing IRRs observed and what this could mean for the incidence of conditions within the population. A crucial limitation of this study that the authors allude to, but do not significantly incorporate at all stages of the interpretation is that the study is looking solely at hospitalisations, not the incidence of disease within the population as a whole. A reduction in hospitalisations for a condition may reflect a higher threshold for hospital admission, not a reduction in population incidence. It is not possible to tease out which of these is happening in this study, and as such, the interpretation should be very guarded.

The implications for this difference are significant from a policy perspective, and this is why guarded interpretation is so important. Suggesting a fall in incidence when what really occurred was a fall in hospitalisation with little if any change in incidence would indicate an increase in untreated conditions / unmet clinical need. One framing of the findings (reduced population incidence) would focus efforts away from a condition, while the other (reduced hospitalisation with unchanged population incidence) would focus efforts towards addressing a potential unmet need.

In the study, all conditions (above an inclusion threshold) are treated with equal importance. This leaves the reader with little understanding of whether conditions occur often or rarely, and how much importance to place on changing incidence. Consider the example of 'haemolytic anaemias' - is this a high volume condition? Are the parameter estimates robust or are margins of error large or findings non-significant. This is potentially also the case for admissions with a primary diagnosis of 'intellectual disabilities', which may be relatively rare, and therefore potentially uncertain estimates. Additionally, in my practice it would be unusual for someone with an intellectual disability to have it as the primary diagnosis code for their admission. Do the authors have any insight as to what these admissions may involve and how their increased rates may be explained. It could for example pertain to admissions arising from a breakdown of homecare or residential care for patients with intellectual disabilities requiring admission elsewhere. I do think the occurrence of 'intellectual disability' as a primary diagnosis code is a bit odd and warrants further investigation or contextualisation.

The extent to which the discussion focusses on these two cases which may be rare in the scope of the overall dataset and whose estimates may be uncertain is perhaps too great. There is less discussion of those conditions in clusters C and D that collectively account for the majority of admissions, and certainly the majority of conditions included.

There are also many instances of overstatement or assertions of the novelty, robustness and uniqueness of the data or the study that I would suggest are either removed or backed up with evidence.

Overall:

The authors have used an interesting dataset to identify some potentially useful findings. I have some significant reservations about the data and the means in which they were analysed and interpreted. I'm unsure of the extent to which some of these can be rectified, however the interpretation of findings should be achievable for the authors, and changes to the reporting of the clustering methods should be also possible. I imagine the limitations of the dataset are unavoidable, however care should be taken to ensure its limitations are adequately expressed in relation to its implications for the study.

Title: *Trends in non-COVID-19 hospitalizations prior to and during the COVID-19 pandemic period, United States, 2017 – 2021*

We kindly thank all three reviewers for their time and consideration in commenting on our manuscript. We truly believe their questions and critiques have improved the quality and depth of our study. Additionally, we would like to show appreciation for the polite phrasing of all critiques which lifted our spirits when reading through the remarks.

REVIEWER COMMENTS

Reviewer #1 (Remarks to the Author):

It has been previously reported that dramatic changes in healthcare utilization have occurred due to the COVID-19 pandemic, including changes in hospitalizations for non-COVID related reasons. This is due to many factors, including pandemic shutdowns, social distancing and face masks, patients avoiding hospitals for screening tests, delays in non-essential procedures, etc. However, the authors state that prior work in this area has focused on individual conditions or smaller populations. This study takes a systematic look at this issue by estimating changes in hospitalizations across all ICD-10 diagnoses for a large U.S. population of 13.5M patients with 19.2M hospitalizations. Diagnoses were grouped into four clusters, with results similar to prior studies. For example, non-COVID infectious diseases in cluster A immediately fell in March 2020 and remained well below normal for the rest of the study period, ending in June 2021.

- **General Comment #1:** Understanding how the pandemic affected non-COVID-19 hospitalizations is essential for many applications. There actually have been other studies like this that looked systematically across diseases for large populations. For example:

In Denmark:

Bodilsen J, Nielsen PB, Søgaard M, Dalager-Pedersen M, Speiser LOZ, Yndigegn T, Nielsen H, Larsen TB, Skjøth F. Hospital admission and mortality rates for non-covid diseases in Denmark during covid-19 pandemic: nationwide population based cohort study. *BMJ*. 2021 May 24;373:n1135. doi: 10.1136/bmj.n1135. PMID: 34035000; PMCID: PMC8142604.

In Spain:

Pifarré I Arolas H, Vidal-Alaball J, Gil J, López F, Nicodemo C, Saez M. Missing Diagnoses during the COVID-19 Pandemic: A Year in Review. *Int J Environ Res Public Health*. 2021 May 17;18(10):5335. doi: 10.3390/ijerph18105335. PMID: 34067807; PMCID: PMC8156815.

In the Philippines:

Uy J, Siy Van VT, Ulep VG, Bayani DB, Walker D. The Impact of COVID-19 on Hospital Admissions for Twelve High-Burden Diseases and Five Common Procedures in the Philippines: A National Health Insurance Database Study 2019-2020. *Lancet Reg Health West Pac.* 2022 Jan;18:100310. doi: 10.1016/j.lanwpc.2021.100310. Epub 2021 Nov 3. PMID: 34751261; PMCID: PMC8565915.

In the US (although not broken out by disease category, this uses a large 8M patient U.S. Medicaid database):

Dang A, Thakker R, Li S, Hommel E, Mehta HB, Goodwin JS. Hospitalizations and Mortality From Non-SARS-CoV-2 Causes Among Medicare Beneficiaries at US Hospitals During the SARS-CoV-2 Pandemic. *JAMA Netw Open.* 2022 Mar 1;5(3):e221754. doi: 10.1001/jamanetworkopen.2022.1754. PMID: 35262712; PMCID: PMC8908076.

Thus, while the systematic look across all diseases is not novel, I do think that the large size of this study adds new accuracy and details that haven't been seen before.

- **General Response #1:** We kindly thank Reviewer 1 for highlighting these studies and have incorporated these as references in both the Introduction and Discussion sections. We feel that our study is still informative, as it uses a larger baseline period to establish and understand pre-pandemic trends, and a larger US population that includes both private and public health insurance.

Major Concerns

Comment #1: A key difference between this study and others is the use of the large COVID-19 Research Database. However, the manuscript says very little about this. Much more detail is needed.

We have answered each subset of this question more specifically below:

- **Comment #1.1)** I am aware of the COVID-19 Research Database, but I know very little about it. My understanding is that it is a collection of databases contributed by different organizations to support COVID-19 research. I think the authors selected a single claims database, Change Healthcare, which combines data from multiple

insurance companies/payers. However, the authors do not actually name the database. Is there a reason for this? Why did they select this database?

- **Response 1.1)** The database used was OfficeAlly and we have added the name of this database to the Methods section. This database was chosen as it was the only database available within the COVID-19 Research Database that contains high-coverage claims data for all patients regardless of cause (rather than just COVID-19 patients as with some other databases).
- **Comment #1.2)** I know a couple of groups that have been trying to use the Change Healthcare data in the COVID-19 Research Database, but they have run into many problems. Can the authors either reference another study that has used this data or provide some information about how they did it (e.g., the process for getting access to the data, what software programs they used for the analysis, etc.). This could be very helpful to others.
 - **Response #1.2)** This analysis used the OfficeAlly data (as is now specified in the Methods section). We have added a reference to previous use of the Office Ally dataset by Zhu et al. (2022, *Health Affairs*) in a study of outpatient mental health services.
 - *“The database can be accessed by academic, scientific and medical researchers conducting real-world data studies related to COVID-19... Study proposals will undergo privacy and governance review, as well as review by the Scientific Steering Committee. Each proposal will be evaluated on the basis of three criteria: (1) the proposal is aligned with the mission of the database, (2) the proposal is feasible, and (3) the individual submitting the proposal is qualified to answer the question proposed”*
 - The software programs used for analysis are described in the Methods section and include Snowflake SQL and R, both of which are accessed through a VPN and AWS.
- **Comment #1.3)** How many different payers are represented in this database? The authors wrote “additional payers were incorporated over time”. Did payers ever leave the database?
 - **Response #1.3)** Only 250 payers left the database between 2017 and June 2021 while 343 payers were added during the same time period.
- **Comment #1.4)** It would be helpful to at least have a summary figure showing (a) the total number of hospitalizations in this database per month, (b) the number of COVID-19 and non-COVID-19 hospitalizations per month, and (c) the number of payers included in the database per month. Seeing the number of payers per month could provide insight into the strange spikes in Jan 2020 and Jan 2021.

- **Response #1.4)** We have added two additional Supplemental Figures to answer these questions. The new Supplemental Figure S1 shows the number of unique payers per month. It shows that the number of payers reporting each month varied from ~750 to ~920 during the study period. The peak ~920 payers per month occurred in Jan 2020, which is why we carefully fit our offset to assume an increasing number of payers through 2020 and early 2021. The new Supplemental Figure S3 shows the total number of non-covid and covid-19 hospitalizations per month and the number of unique payers per month. *These figures are also inserted at the bottom of this response document.* We did not add a separate figure showing the total number of hospitalizations per month as this can be inferred from the new Supplemental Figure S3.
- **Comment #1.5)** What are the age/sex/race/ethnicity distributions in the data?
 - **Response #1.5)** Sex/race/ethnicity information is not available in the OfficeAlly database used for analysis. We have added this as a limitation in the Discussion section. Age information was available but outside of the scope but of interest for further research.
- **Comment #1.6)** The pandemic waves hit different parts of the U.S. at different times, and healthcare utilization was probably affected in different ways by location. For example, masks were used more in some places than others, pandemic shutdowns varied by location, etc. The authors note that their data covered all fifty states and Puerto Rico. However, the distribution is not reported. Because the insurance payers changed over time, the geographic distribution might also have changed. Ideally, the authors could provide some information, at least by region of the country, on coverage by month. If not, then they should comment about this.
 - **Response #1.6)** Unfortunately, it is challenging to match each payer to the exact state or region where care was being provided. We were able to determine that there were payers from all states after analyzing payer names in which state names are specifically included. Other payers, like United Healthcare, Blue Cross Medicare, Blue Shield, etc., do not match to a specific payer state or region. We have clarified this point in the methods section: “Insurance payers covered patients from all fifty states and Puerto Rico but some payers covered patients in multiple states and regions without specifying which state a patient resided in, thus information is not presented by state or region.”
- **Comment #1.7)** The Data Availability statement indicates the data can be obtained from the COVID-19 Research Database consortium. However, I’m not certain I know which subset of that database the authors used for this study. I’m guessing Change Healthcare. Can the authors share the code they used? I’ve heard that it is difficult to write the SQL queries for Change Healthcare. It would be helpful to others to see how they did this. The authors note that they are “in the process of gaining approval...to make the relevant code public.”

- **Response #1.7)** We completely agree and will make both the SQL queries and R code for analysis available on GitHub to aid other researchers using Office Ally within the C19RDB. The database is in the process of approving our updated code for release but they are undergoing a shift in platforms at the moment so we have been told to expect a delay of 1-3 weeks. (The rest of comment #1.7 has been answered above in **Response 1.1 and 1.2**).
- **Comment 2)** It would be good to have a figure like Figure 1, but one graph for each cluster (combining all hospitalizations within the cluster). It is difficult to see the overall trends in Figure 2, where it is broken out by subchapter.
 - **Response #2)** To show overall trends by subchapter of interest, we have plotted timeseries of each individual subchapter (like in Figure 1) for diagnoses in cluster A and cluster B. This Figure is added as **Supplemental Figure S4** and is included at the bottom of this document.
- **Comment 3)** On page 4, line 155, the authors write that they used a “two-stage simulation using Monte Carlo resampling”. They give a couple references, but a brief summary of what this is would be nice.
 - **Response 3)** Agreed, we have added a brief description in the methods. The resampling approach accounts for both parameter uncertainty and observation uncertainty. In the first stage, we resample from a multivariate normal distribution, using the mean and variance-covariance matrix of the regression coefficients. These samples are combined with the covariates to get samples of the regression mean. We then take samples of these samples, where the regression means are drawn from a Poisson distribution. This adds observation uncertainty. The result of this resampling scheme is a 95% prediction interval.

Minor Comments

- **Comment #4)** Figure 2A is for clusters A and B, right? If so, then this should be noted in the caption.
 - **Response #4)** Thank you for pointing this out. We have edited the caption in Figure 2A to clarify this is referencing clusters A and B. We have also darkened the second Y-axis to make the labeling legible.

- **Comment #5)** In Figure S1, the de-duplication step increased the number of records from 33.6M to 34.1M. Is this a typo? How did removing duplicates increase the number?
 - **Response #5)** Thank you for bringing this to our attention. We have corrected the figure and added more detail to the flowchart so that it is clearer how many individuals are leaving at each stage of the data cleaning process.
- **Comment #6)** Is the caption for Figure S3 correct?
 - **Response #6)** This caption was our (embarrassing) mistake and has since been fixed. Thank you for noting this.

Reviewer #2 (Remarks to the Author):

General comments:

This is an interesting epidemiological study of the estimated effects of the COVID-19 pandemic on non-COVID-19 hospitalization trends between the beginning of the pandemic and June 2021. The use of a large database with 3 years of hospitalization data prior to the start of the pandemic combined with a thorough analysis results in a convincing depiction of these trends.

We thank the reviewer for their positive review of our work.

- **General Comment A)** The most significant limitation is the lack of data beyond June 2021 – which means that there is no information on trends during the Delta and Omicron waves, which were both extremely significant, but also occurred in a population who had been living with the pandemic for over a year already (see attached image file). In addition, vaccination began in December 2020, and as a result most of this study’s period of interest is in the pre-vaccination period, which would obviously have a huge impact on population-level behavior.
 - **Response A):** Unfortunately, the last update to this database was provided in September 2021 and C19RDB informed us that there is a potential 3-month lag in reporting, we decided to err on the side of caution and end our analysis in June 2021. We realize that this excludes a large proportion of the most recent epidemic waves but are limited by the database itself. We have a statement in the Methods section to indicate that study length is due to data limitations: “Reporting to this database was continually updated until September 2021...”

- **General Comment B)** The result of this limitation may be that this manuscript does not hold the practical utility the authors suggest, and the focus should be more on the interesting epidemiological findings of the detailed effects of a large pandemic on non-COVID hospital services.

Furthermore, due to the methodology and types of data used, it is difficult to ascertain causality: a whole range of things changed with the onset of the pandemic and teasing apart which factor influence the drop is not possible from this type of analysis. As such, once again, the language could be focused more on describing the epidemiological findings than on possible causative mechanisms (specific points in the comments below).

- **Response B):** We have tailored the manuscript according to the recommendations below so that the practical utility for other researchers and clinicians remains true. We feel this study offers useful insight for fellow researchers who are interested in key trends but can only access smaller datasets and thus cannot draw immediate conclusions on whether the trends they see may be meaningful for further exploration.
- Language used in the Results and Discussion section is careful to describe association and not causation. We do not imply that any of these findings show causal mechanisms and take care to offer alternative explanations for certain findings in the Discussion section. We have further edited the manuscript to ensure causal language is avoided and secular trends in the data are described as needed. The discussion section attempts to contextualize the findings within the broader literature to better understand potential mechanisms at play; however, this does not imply causality.
- Numerous studies have demonstrated that the effects of the COVID-19 pandemic on healthcare utilization were most severe in the early phases of the pandemic. Our study of this critical period of the pandemic can be important to understand the short- and long-term consequences of these shifts in hospitalization trends.

Detailed comments:

Introduction:

- **Comment 1 Lines 72-75:** The two articles referenced here were published at a much earlier stage in the pandemic. Do the results of the current manuscript (that show trends up until June 2021 – before Delta and Omicron and in a predominantly pre-vaccination period) really aid us in hospital preparation and resource allocation? If so, how?
 - **Response 1 Lines 72-75:** We see Reviewer 2's point that this analysis does not capture the most recent waves of the pandemic and have removed this sentence so as to not overstate our impact. We still feel this study is important because it covers the early pandemic period, prior to restructuring to

accommodate capacity issues. This study is more specifically about disruptions during and after the unique crisis period at the start of the pandemic.

- **Comment 2 Lines 79-81:** The analysis results are limited in their ability to inform such specific hypotheses – in order to inform the hypothesis for example that NPIs influenced hospitalizations numerous confounders would have to be dealt with, and this is out of the scope of the current article. Consider removing this sentence.
 - **Response 2 Lines 79-81:** We agree with Reviewer 2 on this point and have removed this sentence. Instead we focus on our ability to leverage multiple years of pre-pandemic data to aid in our estimations, the incorporations of public and private insurance patient data from the US and the ability to estimate changes in both common and uncommon causes of hospitalization, thereby being of interest to a range of specialties.

Methods:

- **Comment 3 Lines 114-118:** Does this include primary as well as secondary diagnostic codes? The number of hospitalizations excluded (66,099) seems a little small on the basis that we are drawing from a population significantly larger than 13.5 million individuals who were hospitalized in this dataset. From the CDC website it appears as if about nearly 2.5 million hospitalizations occurred in the US by June 2021 (<https://www.cdc.gov/coronavirus/2019-ncov/covid-data/covidview/index.html>). Maybe some of the hospitalizations removed because they occurred within a 30-day window of prior hospitalization were COVID-19 related?
 - **Response 3 Lines 114-118:** We have reformatted the Supplemental Figure (now S2) that shows the data inclusion/exclusion process to clarify this point. Previously the COVID-19 cases were shown as being excluded later in the process and thus looked erroneously small. You are correct, some of the COVID hospitalizations were removed in the 30d-window of hospitalization.
- **Comment 4 Line 133:** Please note that in Supplementary Table S2 it is written that subchapters with fewer than 500 diagnoses (rather than 5000) were excluded.
 - **Response 4 Line 133:** Thank you for pointing this out! We have fixed this mistake in the revised draft.

Overall, the development of the Poisson models to predict the expected rates based on the previous three years seem to have been performed rigorously and provides a good estimated baseline for expected rates.

- **Response:** *We kindly thank the reviewer for this positive feedback*

Results:

- **Comment 5 Lines 171-177:** The phrasing here is confusing. In general the monthly unique hospitalizations including the relevant trends should be described from Jan 2017 until Feb 2020 (including an additional mention of a large increase in Jan and Feb 2020 if indeed there was a big jump to 455,000 from less than 400,000 in any month prior to 2020) and then a description of the general trends from March 2020 until June 2021. This information is not presented in a figure or table (at least no figure or table is referenced) and as such needs to be clearly stated here in full. It might be also useful to note here the number of excluded COVID-19 related hospitalizations during the pandemic period.
 - **Response 5 Lines 171-177:** We chose to report the difference in hospitalizations as such to highlight the secular changes in the database between Dec 2019 and Jan 2020. In this revision, we have added a supplemental figure (**Supplemental Figure S3** inserted at the bottom of this document) to show the number of monthly all-cause hospitalizations (primary diagnostic code field only), and monthly COVID-19 hospitalizations (all diagnostic code fields).
- **Comment 6 Lines 181-183:** It is confusing to report relative increases across all groups and then present an IRR of 0.96 which would represent a decrease. Consider specifically commenting on Conditions in the perinatal period as the only chapter that didn't show an increase.
 - **Response 6 Lines 181-183:** We have incorporated this feedback and rephrased those sentences as such: *“All ICD-10 chapters experienced relative increases in incidence of hospitalizations during the pre-pandemic months of January and February 2020, with IRRs ranging from 1.08 to 1.21 in Jan 2020 (**Figure 1; Supplemental Table S3**), likely reflecting database reporting changes at the start of the new calendar year. The exception being conditions of the perinatal period, which had an estimated IRR of 0.96 (95%CI: 0.87, 1.06) in Jan 2020.”*
- **Comment 7 Lines 184-185:** Likewise with March 2020, the perinatal period chapter is the same in March as February and as such this sentence is confusing.
 - **Response 7 Lines 184-185:** The relative decline reported in March 2020 is compared to previous years data (March 2019, March 2018, March 2017), not Feb 2020 data. Thus, conditions of the perinatal does still show a relative decline compared to March 2019. We did alter the statement to be *“Beginning in March 2020, the relative incidence of hospitalization among all diagnostic chapters exhibited a moderate decrease compared to baseline.”* to clarify this point.
- **Comment 8 Lines 196-243:** The description of the clusters is a bit clunky and could be streamlined. Maybe consider leading with the types of conditions in each cluster

followed by the trends of that cluster of the pandemic. For example: Cluster A consists of the subchapters [list relevant subchapters]. These subchapters demonstrated a sharp relative decrease in hospitalizations early in the pandemic with rates below expected until [the end of 2020? June 2021?]. Then provide some specific details of interest such as rates of P&I. It may help with the flow and clarity of the clusters.

- **Response 8 Lines 196 – 243:** Thank you for this advice. We have altered the Results section to introduce the subchapters first and then trends second as suggested.
- **Comment 9 Lines 218-225:** Specifically, with cluster B, the phrasing is confusing – consider “the average IRR in this cluster remained at or below expected levels from March 2020 – May 2020 and exceeded expected levels for the remainder of the study period.”
 - **Response 9 Lines 218-225:** We appreciate this feedback and have incorporated the suggested phrasing.
- **Comment 10 Lines 242-243:** Please update to ‘between March 2020 and June 2021. Current phrasing ‘between March and June 2021’ is confusing
 - **Response 10 Lines 242-243:** We see how this is confusing and thank you for pointing this out. We have changed the sentence as suggested.

Discussion:

- **Comment 11 Lines 276-318:** Generally, this section needs to be cut down. A significant proportion of the text is spent trying to explain what seem like possible artefacts in the data or things that the data cannot explain such as the trends in hospitalizations where the primary diagnosis is intellectual disabilities. It might be more effective to build a paragraph around ‘signals’ generated that could be leads for further research (is there a causal relationship between SARS-CoV-2 pandemic and hemolytic anemias/intellectual disabilities/adverse newborn outcomes). In addition, as a way of providing some validation to the results, it would be interesting to show that the number of childbirth coded diagnoses (i.e. sub-chapter) didn’t drop in the early pandemic (it should be unaffected overall as the authors mention) and that the drop in the pregnancy, childbirth and puerperium diagnoses occurred as a result of codes from the pregnancy and puerperium related codes.
 - **Response 11 Lines 276-318:** The discussion section has been edited to remove the paragraphs explaining relatively minor trends in the data (diagnostic codes for psychological disturbances) and to shift focus to the more impactful trends. Trends that we cannot readily explain (since the dataset doesn’t allow for interpretations of mechanisms), we have provided references for further reading but have limited our focus. Figure 2B shows

that maternal delivery codes were not significantly altered early in the pandemic period, in contrast to the codes for newborn gestation disorder codes.

Conclusion:

- **Comment 12 Lines 242-243** Lines 380-387: The conclusion here focusses a lot on the aspects that the analysis cannot actually answer – why the trends are what they are. An alternative focus for the conclusion could be that for the first time, by leveraging a large billing database in the US, a comprehensive high-level analysis of hospitalization by diagnosis has identified broad trends over the pandemic period. This is useful because...[insert reasons].
 - **Response 12 Lines 242-243:** We have incorporated this suggestion throughout the discussion section and switched the focus on aspects of why the analysis is important, rather than signals in the data that are challenging to explain.
- **Comment 13 Lines 387-388:** It is unclear if the last sentence adds value to the manuscript and if so, how this will help minimize disruptions during future major health events. Recommend removing and leaving the reader to decide if the findings are of interest/use.
 - **Response 13 Lines 387-388:** We have removed the mentioned sentence as suggested.

Miscellaneous:

Supplementary figure S3 – The legend seems to be unedited: “shows the clusters of the subchapters and where I made the cut-offs – can update this to have clear demarcations of the cut-offs” – please correct.

- **Response:** Apologies, we have fixed this mistake in this revision. Thank you for highlighting this.

Reviewer #3 (Remarks to the Author):

Thank you for giving me the opportunity to review this manuscript examining changing rates of hospitalisation for a range of conditions in the USA between 2017 and 2021.

The article uses an administrative healthcare claims dataset which represents a non-random sample of data provided by a range of primary and secondary healthcare institutions. The authors use these data to estimate monthly incidence rate ratios for hospitalisation for a panel of conditions derived from the ICD-10 codes of the primary

diagnosis of an admission. The authors then use hierarchical clustering to identify four clusters of conditions whose changing incidence rate ratios are similar to one another.

This is an interesting study that leverages a large dataset to look at month to month changes in incidence, in addition to looking at a large number of conditions. The differences in trajectories for the clusters identified are interesting and may provide an important insight into the ways health systems provided care to patients with different conditions.

Response: *We kindly thank the reviewer for their positive feedback.*

I do have several concerns regarding the dataset, methods used and the way in which findings are interpreted.

- **Comment 1.** The underlying dataset used.

The authors acknowledge that the C19RDB dataset is imperfect, relying on the contribution of claims data from providers over time and resulting in sometimes large fluctuations in the amount of data received. It is not clear the extent to which these fluctuations are random with respect to the underlying features of the data received, whether with respect to centres providing data and the characteristics of their patients. This also leads to a lack of a population denominator for their models, which the authors acknowledge. It would be useful to understand the sensitivity of their models to changes in the assumptions of the active patient population (lines 146-149).

- **Response 1:** We acknowledge Reviewer 3's concerns and to address this, we conducted a resampling analysis that randomly selected 20 of the top 100 largest payers to remove from analysis prior to calculating the IRR using only those providers. We repeated this 1000 times and then reported the median and 95% sample intervals in the supplemental text. We show that are results are robust to the mix of providers reporting in the database.

- **Comment 2.** Clustering methodology.

The article relies heavily on the identification of clusters of conditions and the discussion consists mostly of discussion of these clusters and their collective temporal trends. As such, I would hope to have far more detailed explanation of the clustering process used, considerations made on methods used and how data were prepared prior to clustering. Was each month treated as an independent data point, or were efforts made to otherwise characterise the time series in a higher order manner by relating each data point to its neighbours or similar? Why was hierarchical clustering chosen over other methods? Not to say it's a bad choice per se, but the authors also use K-means clustering alongside and then choose to adopt the clustering derived from a hierarchical method.

I'm not sure about the appropriateness of using K-means clustering with an elbow

plot to justify the choice of cluster number for a hierarchical clustering output. Both are different methods and may have different optimal partitions. The dendrogram should be enough to determine which of many possible configurations is notionally optimal for the purpose of the analysis.

There is no clearly optimal clustering configuration to the data, as there rarely are outside of purely artificial datasets. That said, looking at the PCA plot, it seems like the 4 cluster configuration is perhaps inappropriately splitting the central mass of the point cloud. This is perhaps backed up by Figure S3 in which either 3 or 7 clusters looks to be the best partition of the heat map. The elbow plot is a pretty gentle transition, with barely an elbow at all, so I'm not sure it strongly supports a choice of 4 clusters. 3 clusters is perhaps the right split to avoid overselling difference observed between trajectories.

Given the central role of the clustering process to the paper, I would expect more of the clustering figures to feature in the main dataset, e.g. Figure S3.

- **Response 2:** Yes, we thought there could be a split at either 3 or 4 clusters and used k-means as a second form of validation; however, we see your point that one method should be preferred. We chose hierarchical clustering because it is readily interpretable and does not rely on a prior expectations of the number of clusters. We are happy to err on the side of caution and report only 3 clusters. This does not drastically alter the results because previously named clusters C and D would be considered one cluster under a 3-cluster split. Figure 2B and the Results section have been updated to reflect this change. We have also simplified the description of our methods in the Methods section to only reflect the hierarchical clustering approach.
- **Comment 3.** Interpretation of findings.

In the discussion, the authors explore the changing IRRs observed and what this could mean for the incidence of conditions within the population. A crucial limitation of this study that the authors allude to, but do not significantly incorporate at all stages of the interpretation is that the study is looking solely at hospitalisations, not the incidence of disease within the population as a whole. A reduction in hospitalisations for a condition may reflect a higher threshold for hospital admission, not a reduction in population incidence. It is not possible to tease out which of these is happening in this study, and as such, the interpretation should be very guarded.

- **Response 3:** This is an important limitation and we have edited the Strengths/Limitations section to incorporate this statement “Similarly, we are unable to make inferences on the prevalence or incidence of disease in the general population, since the threshold for hospitalization may have changed during the pandemic period”

- **Comment 4.** The implications for this difference are significant from a policy perspective, and this is why guarded interpretation is so important. Suggesting a fall in incidence when what really occurred was a fall in hospitalisation with little if any change in incidence would indicate an increase in untreated conditions / unmet clinical need. One framing of the findings (reduced population incidence) would focus efforts away from a condition, while the other (reduced hospitalisation with unchanged population incidence) would focus efforts towards addressing a potential unmet need.
 - **Response 4:** We have adjusted our Results and Discussion sections to highlight that we are referring to relative incidence and incidence of hospitalization, not population level incidence. Additionally, we have reflected this notion in the limitations section of the Discussion: “Similarly, we are unable to make inferences on the prevalence or incidence of disease in the general population, since the threshold for hospitalization may have changed during the pandemic period and hospitalization rates of disease rarely match incidence rates of disease in the general population”
- **Comment 5.** In the study, all conditions (above an inclusion threshold) are treated with equal importance. This leaves the reader with little understanding of whether conditions occur often or rarely, and how much importance to place on changing incidence. Consider the example of 'haemolytic anaemias' - is this a high volume condition? Are the parameter estimates robust or are margins of error large or findings non-significant. This is potentially also the case for admissions with a primary diagnosis of 'intellectual disabilities', which may be relatively rare, and therefore potentially uncertain estimates. Additionally, in my practice it would be unusual for someone with an intellectual disability to have it as the primary diagnosis code for their admission. Do the authors have any insight as to what these admissions may involve and how their increased rates may be explained. It could for example pertain to admissions arising from a breakdown of homecare or residential care for patients with intellectual disabilities requiring admission elsewhere. I do think the occurrence of 'intellectual disability' as a primary diagnosis code is a bit odd and warrants further investigation or contextualisation.
 - **Response 5:** Supplementary Table S4 provides IRR and 95% prediction intervals for these specific diagnoses that are seemingly odd/rare. We have added examples of the IRR (95%PI) to the text to make this information more readily available to the reader to interpret. We have also added a time series plot, similar to Figure 1, for these conditions of interest to show the absolute numbers of hospitalizations per month and condition. We are unsure why “intellectual disabilities” would be a primary DX code but include a statement that it could be related to “infection or other pandemic-related changes in care for members of this population”. We have added supporting references to this

statement. We feel that this topic has been contextualized and alternative hypotheses presented in this revision.

- **Comment 6:** The extent to which the discussion focusses on these two cases which may be rare in the scope of the overall dataset and whose estimates may be uncertain is perhaps too great. There is less discussion of those conditions in clusters C and D that collectively account for the majority of admissions, and certainly the majority of conditions included.
 - **Response 6:** We appreciate this feedback. We have edited the discussion section to focus more on the conditions that account for the majority of admissions.
- **Comment 7:** There are also many instances of overstatement or assertions of the novelty, robustness and uniqueness of the data or the study that I would suggest are either removed or backed up with evidence.
 - **Response 7:** We have removed statements regarding resource allocation, hospital preparation and disease burden, so as to not overstate our findings. We feel the novelty lies in the breadth of the database, its ability to capture cases using both private and public insurance, and the extra years of pre-pandemic comparison data to improve our estimates of IRR. We hope that specialists and generalist physicians alike, as well as epidemiologists, can find utility in this study when analyzing trends in their own data.

Overall:

The authors have used an interesting dataset to identify some potentially useful findings. I have some significant reservations about the data and the means in which they were analysed and interpreted. I'm unsure of the extent to which some of these can be rectified, however the interpretation of findings should be achievable for the authors, and changes to the reporting of the clustering methods should be also possible. I imagine the limitations of the dataset are unavoidable, however care should be taken to ensure its limitations are adequately expressed in relation to its implications for the study.

Supplemental Figure S1. Number of unique payers in the dataset per month, Jan 1, 2017 – Jun 31, 2021

Supplemental Figure S3. Number of non-COVID-19 hospitalizations (in ICD-10 chapters A-T) and COVID-19 hospitalizations (all diagnostic field codes) analyzed, Jan 1, 2017 – Jun 31, 2021

Trends in non-COVID-19 hospitalizations; NCOMMS-22-16251

Supplemental Figure S6. Time series of observed hospitalizations per diagnostic subchapter by month and year (solid line) with model predicted case counts (dashed line) and confidence intervals (grey).

Trends in non-COVID-19 hospitalizations; NCOMMS-22-16251

REVIEWERS' COMMENTS:

Reviewer #1 (Remarks to the Author):

I think this manuscript is much improved over the previous version. The authors addressed my concerns, especially with clarifying the source of the data and adding figures that give better context to the results (e.g., total number of payers and hospitalizations per month). Changes based on comments from the other reviewers, such as using only three clusters and modifying the focus on how the results are described, also make this much easier to read.

Minor comment:

1) Line 185 states that the monthly hospitalizations from 2017 to 2019 ranged from ~300,000 to ~400,000. However, in the new Figure S3, it looks like the lower bound should be ~200,000 rather than ~300,000.

I don't think it is necessary for the authors to address this, but I thought it was interesting that in Figure S3, in each year from 2017 to 2019, there is a spike in hospitalizations in January. The counts fall throughout the first half of the year until they settle at about 50%-70% the January level. Much of this is driven by normal seasonal infectious disease patterns. Though, some of it might also be due to how OfficeAlly is constructed, or because of new insurance policies beginning in January. Line 186 states that hospitalizations from January to April 2020 dropped from 455,000 to 151,000. That sounds huge, but actually a lot of that is just the normal January to April drop seen in each of the previous three years. Of course, the analysis corrects for this and the rest of the results are presented as deviations from the expected counts. So, this isn't a problem. However, until the authors added Figure S3, I thought there was a far greater drop than there truly was. It is also interesting that in Figure S1, from 2019 to 2021, January 2020 has the most payers and April 2020 had the fewest. This also might have exaggerated the drop a bit over that four month period.

Reviewer #2 (Remarks to the Author):

** Reviewer Responses: 1 for each general comment and then an additional response for each of the introduction, methods, results and discussion sections.

General comments:

This is an interesting epidemiological study of the estimated effects of the COVID-19 pandemic on non-COVID-19 hospitalization trends between the beginning of the pandemic and June 2021. The use of a large database with 3 years of hospitalization data prior to the start of the pandemic combined with a thorough analysis results in a convincing depiction of these trends.

We thank the reviewer for their positive review of our work.

** Reviewer Response: The revised article reads well and the authors have taken care to respond to the comments. I have a few further comments in response which can be found below.

General Comment A) The most significant limitation is the lack of data beyond June 2021 – which means that there is no information on trends during the Delta and Omicron waves, which were both extremely significant, but also occurred in a population who had been living with the pandemic for over a year already (see attached image file). In addition, vaccination began in December 2020, and as a result most of this study's period of interest is in the pre-vaccination period, which would obviously have a huge impact on population-level behavior.

Response A): Unfortunately, the last update to this database was provided in September 2021 and

C19RDB informed us that there is a potential 3-month lag in reporting, we decided to err on the side of caution and end our analysis in June 2021. We realize that this excludes a large proportion of the most recent epidemic waves but are limited by the database itself. We have a statement in the Methods section to indicate that study length is due to data limitations: "Reporting to this database was continually updated until September 2021..."

** Reviewer Response: I understood that the limitation was a data limitation – but this does not prevent it from being a limitation. Whilst this addition makes it clear that the study length is due to data limitations, it would be good to clarify in the introduction that the study aims to assess the relative changes in the incidence of hospitalizations by diagnostic category in the early-medium phase (first 18 months) of the COVID-19 pandemic. I think it may be prudent to also add in a sentence in the limitations section given that this clearly limits the scope of the interpretation of the results. In general, the framing of the clinical utility has been improved and as such this is less problematic in this version of the manuscript.

General Comment B) The result of this limitation may be that this manuscript does not hold the practical utility the authors suggest, and the focus should be more on the interesting epidemiological findings of the detailed effects of a large pandemic on non-COVID hospital services. Furthermore, due to the methodology and types of data used, it is difficult to ascertain causality: a whole range of things changed with the onset of the pandemic and teasing apart which factor influence the drop is not possible from this type of analysis. As such, once again, the language could be focused more on describing the epidemiological findings than on possible causative mechanisms (specific points in the comments below).

Response B): We have tailored the manuscript according to the recommendations below so that the practical utility for other researchers and clinicians remains true. We feel this study offers useful insight for fellow researchers who are interested in key trends but can only access smaller datasets and thus cannot draw immediate conclusions on whether the trends they see may be meaningful for further exploration.

Language used in the Results and Discussion section is careful to describe association and not causation. We do not imply that any of these findings show causal mechanisms and take care to offer alternative explanations for certain findings in the Discussion section. We have further edited the manuscript to ensure causal language is avoided and secular trends in the data are described as needed. The discussion section attempts to contextualize the findings within the broader literature to better understand potential mechanisms at play; however, this does not imply causality.

Numerous studies have demonstrated that the effects of the COVID-19 pandemic on healthcare utilization were most severe in the early phases of the pandemic. Our study of this critical period of the pandemic can be important to understand the short- and long-term consequences of these shifts in hospitalization trends.

** Reviewer Response: The results and discussion sections read better now, and contextualise well without implying causation. The language regarding the practical utility of the manuscript is now more appropriately focussed. A few minor detailed points are added below.

Detailed comments:

Introduction:

Comment 1 Lines 72-75: The two articles referenced here were published at a much earlier stage in the pandemic. Do the results of the current manuscript (that show trends up until June 2021 – before Delta and Omicron and in a predominantly pre-vaccination period) really aid us in hospital preparation and resource allocation? If so, how?

Response 1 Lines 72-75: We see Reviewer 2's point that this analysis does not capture the most recent waves of the pandemic and have removed this sentence so as to not overstate our impact.

We still feel this study is important because it covers the early pandemic period, prior to restructuring to accommodate capacity issues. This study is more specifically about disruptions during and after the unique crisis period at the start of the pandemic.

Comment 2 Lines 79-81: The analysis results are limited in their ability to inform such specific hypotheses – in order to inform the hypothesis for example that NPIs influenced hospitalizations numerous confounders would have to be dealt with, and this is out of the scope of the current article. Consider removing this sentence.

Response 2 Lines 79-81: We agree with Reviewer 2 on this point and have removed this sentence. Instead we focus on our ability to leverage multiple years of pre-pandemic data to aid in our estimations, the incorporations of public and private insurance patient data from the US and the ability to estimate changes in both common and uncommon causes of hospitalization, thereby being of interest to a range of specialties.

** Reviewer Response: Overall the introduction has improved. The only phrase that still jars is the phrasing “progressed through multiple waves of SARS-CoV2 variants”, which still in some way overstates the scope of the study. It would be more appropriate to phrase it “progressing through the first two major waves of SARS-CoV2 variants” or “and progressed through the first 18 months of the pandemic”.

Methods:

Comment 3 Lines 114-118: Does this include primary as well as secondary diagnostic codes? The number of hospitalizations excluded (66,099) seems a little small on the basis that we are drawing from a population significantly larger than 13.5 million individuals who were hospitalized in this dataset. From the CDC website it appears as if about nearly 2.5 million hospitalizations occurred in the US by June 2021 (<https://www.cdc.gov/coronavirus/2019-ncov/covid-data/covidview/index.html>). Maybe some of the hospitalizations removed because they occurred within a 30-day window of prior hospitalization were COVID-19 related?

Response 3 Lines 114-118: We have reformatted the Supplemental Figure (now S2) that shows the data inclusion/exclusion process to clarify this point. Previously the COVID-19 cases were shown as being excluded later in the process and thus looked erroneously small. You are correct, some of the COVID hospitalizations were removed in the 30d-window of hospitalization.

Comment 4 Line 133: Please note that in Supplementary Table S2 it is written that subchapters with fewer than 500 diagnoses (rather than 5000) were excluded.

Response 4 Line 133: Thank you for pointing this out! We have fixed this mistake in the revised draft. Overall, the development of the Poisson models to predict the expected rates based on the previous three years seem to have been performed rigorously and provides a good estimated baseline for expected rates.

Response: We kindly thank the reviewer for this positive feedback

** Reviewer Response: The first half of the new flow chart is quite confusing. It might be clearer if those with missing diagnostic info are removed initially and then those with COVID-19 hospitalizations and then those with a hospitalization within 30d of previous. Other than that I have no further comments on the methods section.

Results:

Comment 5 Lines 171-177: The phrasing here is confusing. In general the monthly unique hospitalizations including the relevant trends should be described from Jan 2017 until Feb 2020 (including an additional mention of a large increase in Jan and Feb 2020 if indeed there was a big jump to 455,000 from less than 400,000 in any month prior to 2020) and then a description of the

general trends from March 2020 until June 2021. This information is not presented in a figure or table (at least no figure or table is referenced) and as such needs to be clearly stated here in full. It might be also useful to note here the number of excluded COVID-19 related hospitalizations during the pandemic period.

Response 5 Lines 171-177: We chose to report the difference in hospitalizations as such to highlight the secular changes in the database between Dec 2019 and Jan 2020. In this revision, we have added a supplemental figure (Supplemental Figure S3 inserted at the bottom of this document) to show the number of monthly all-cause hospitalizations (primary diagnostic code field only), and monthly COVID-19 hospitalizations (all diagnostic code fields).

Comment 6 Lines 181-183: It is confusing to report relative increases across all groups and then present an IRR of 0.96 which would represent a decrease. Consider specifically commenting on Conditions in the perinatal period as the only chapter that didn't show an increase.

Response 6 Lines 181-183: We have incorporated this feedback and rephrased those sentences as such: "All ICD-10 chapters experienced relative increases in incidence of hospitalizations during the pre-pandemic months of January and February 2020, with IRRs ranging from 1.08 to 1.21 in Jan 2020 (Figure 1; Supplemental Table S3), likely reflecting database reporting changes at the start of the new calendar year. The exception being conditions of the perinatal period, which had an estimated IRR of 0.96 (95%CI: 0.87, 1.06) in Jan 2020."

Comment 7 Lines 184-185: Likewise with March 2020, the perinatal period chapter is the same in March as February and as such this sentence is confusing.

Response 7 Lines 184-185: The relative decline reported in March 2020 is compared to previous years data (March 2019, March 2018, March 2017), not Feb 2020 data. Thus, conditions of the perinatal does still show a relative decline compared to March 2019. We did alter the statement to be "Beginning in March 2020, the relative incidence of hospitalization among all diagnostic chapters exhibited a moderate decrease compared to baseline." to clarify this point.

Comment 8 Lines 196-243: The description of the clusters is a bit clunky and could be streamlined. Maybe consider leading with the types of conditions in each cluster followed by the trends of that cluster of the pandemic. For example: Cluster A consists of the subchapters [list relevant subchapters]. These subchapters demonstrated a sharp relative decrease in hospitalizations early in the pandemic with rates below expected until [the end of 2020? June 2021?]. Then provide some specific details of interest such as rates of P&I. It may help with the flow and clarity of the clusters.

Response 8 Lines 196 – 243: Thank you for this advice. We have altered the Results section to introduce the subchapters first and then trends second as suggested.

Comment 9 Lines 218-225: Specifically, with cluster B, the phrasing is confusing – consider "the average IRR in this cluster remained at or below expected levels from March 2020 – May 2020 and exceeded expected levels for the remainder of the study period."

Response 9 Lines 218-225: We appreciate this feedback and have incorporated the suggested phrasing.

Comment 10 Lines 242-243: Please update to 'between March 2020 and June 2021. Current phrasing 'between March and June 2021' is confusing

Response 10 Lines 242-243: We see how this is confusing and thank you for pointing this out. We have changed the sentence as suggested.

** Reviewer Response: Overall the results section reads well and is easier to follow. Regarding supplemental figure S3 I would recommend adding in a vertical line at January of each year so that the changes during the period of interest are clearly and it would be easier to gauge how much things changed year to year. Regarding comment 7 – my mistake.

Discussion:

Comment 11 Lines 276-318: Generally, this section needs to be cut down. A significant proportion of the text is spent trying to explain what seem like possible artefacts in the data or things that the data cannot explain such as the trends in hospitalizations where the primary diagnosis is intellectual disabilities. It might be more effective to build a paragraph around 'signals' generated that could be leads for further research (is there a causal relationship between SARS-CoV-2 pandemic and hemolytic anemias/intellectual disabilities/adverse newborn outcomes). In addition, as a way of providing some validation to the results, it would be interesting to show that the number of childbirth coded diagnoses (i.e. sub-chapter) didn't drop in the early pandemic (it should be unaffected overall as the authors mention) and that the drop in the pregnancy, childbirth and puerperium diagnoses occurred as a result of codes from the pregnancy and puerperium related codes.

Response 11 Lines 276-318: The discussion section has been edited to remove the paragraphs explaining relatively minor trends in the data (diagnostic codes for psychological disturbances) and to shift focus to the more impactful trends. Trends that we cannot readily explain (since the dataset doesn't allow for interpretations of mechanisms), we have provided references for further reading but have limited our focus. Figure 2B shows that maternal delivery codes were not significantly altered early in the pandemic period, in contrast to the codes for newborn gestation disorder codes.

** Reviewer Response: Overall the discussion reads better, with an improved focus to the major trends and reduced focus on minor trends with unclear mechanisms that cannot really be teased apart with this dataset. The sensitivity analysis added is a good way of mitigating bias introduced by large payers and contributes to the robustness of the results presented.

Conclusion:

Comment 12 Lines 242-243 Lines 380-387: The conclusion here focusses a lot on the aspects that the analysis cannot actually answer – why the trends are what they are. An alternative focus for the conclusion could be that for the first time, by leveraging a large billing database in the US, a comprehensive high-level analysis of hospitalization by diagnosis has identified broad trends over the pandemic period. This is useful because...[insert reasons].

Response 12 Lines 242-243: We have incorporated this suggestion throughout the discussion section and switched the focus on aspects of why the analysis is important, rather than signals in the data that are challenging to explain.

Comment 13 Lines 387-388: It is unclear if the last sentence adds value to the manuscript and if so, how this will help minimize disruptions during future major health events. Recommend removing and leaving the reader to decide if the findings are of interest/use.

Response 13 Lines 387-388: We have removed the mentioned sentence as suggested.

** Reviewer Response: Overall the conclusion reads well and more accurately represents the findings of the study and its utility.

Miscellaneous:

Supplementary figure S3 – The legend seems to be unedited: "shows the clusters of the subchapters and where I made the cut-offs – can update this to have clear demarcations of the cut-offs" – please correct.

Response: Apologies, we have fixed this mistake in this revision. Thank you for highlighting this.

Reviewer #3 (Remarks to the Author):

Thank you to the authors for the extent to which they have incorporated suggestions from my prior review into their manuscript.

The manuscript reads much better and is a clearer and more balanced article. There are a few points remaining which I would encourage the authors to consider to enhance the clarity of the paper.

Further clarity for the reader as to what admissions with a primary diagnosis of 'intellectual disabilities', 'physiological disturbances' and 'biomechanical lesions' actually mean would also be very helpful. These are often quite broad or vague definitions, and the latter two are rarely used in my everyday practice at least! I still think it would be very helpful to mention the incidence of hospitalisation in each group to give the readers an idea of whether they constitute low or high volume conditions.

If anything, one of the most useful figures in the whole manuscript is figure S6 (in conjunction with Figure 1). This is the only place in the whole manuscript where I get an idea of the relative incidence of hospitalisations in each smaller diagnostic group. It shows me that many of the conditions referred to in the manuscript proper are rare (biomechanical lesions, intellectual disabilities, haemolytic anaemias, physiological disturbances) in comparison to the other diagnoses. Given these small numbers, the potential for statistical noise is high. There are higher volume conditions (such as URI and LRI, middle ear, mental disorders) that show profound trends that may be more robust to describe.

From the perspective of a policymaker, it is smaller fluctuations in more common conditions that are perhaps of most interest, rather than larger fluctuations in less common conditions. In the case of these rarer conditions, it is likely that this sort of 'catch all' administrative data study is not the best way to evaluate changes in such conditions and as such they should not occupy the volume of results and discussion they do.

****Minor points:****

I think it's 'genital tract' rather than 'genital track'.

Looking at the colormaps, the centre point of the map (0.90-1.10) is yellow, and this alludes to the positive end of the scale, and perhaps suggests to the reader values greater than one. Perhaps have this centre value be white / grey so as to avoid confusion here.

Overall, the manuscript is clearer and more focussed, however I do have concerns about the emphasis placed on less common conditions where there may be greater uncertainty about the estimates obtained and trends observed.

Title: *Trends in non-COVID-19 hospitalizations prior to and during the COVID-19 pandemic period, United States, 2017 – 2021*

We kindly thank all three reviewers again for their time and consideration in reviewing multiple versions of our manuscript in an effort to improve the quality. We believe this final version is the strongest version of the manuscript and that it addresses all reviewer comments to the best of our ability. Newer responses are colored blue to distinguish from previous comment/response strings.

REVIEWER COMMENTS

Reviewer #1 (Remarks to the Author):

I think this manuscript is much improved over the previous version. The authors addressed my concerns, especially with clarifying the source of the data and adding figures that give better context to the results (e.g., total number of payers and hospitalizations per month). Changes based on comments from the other reviewers, such as using only three clusters and modifying the focus on how the results are described, also make this much easier to read.

Thank you very much for the insightful comments and the time taken to review multiple versions of our manuscript. We are pleased to hear that the manuscript has improved.

Minor comment:

1) Line 185 states that the monthly hospitalizations from 2017 to 2019 ranged from ~300,000 to ~400,000. However, in the new Figure S3, it looks like the lower bound should be ~200,000 rather than ~300,000.

Response 1) *Thank you for pointing this out. This was referring to unique hospitalizations which approximates unique payers per year, but we realize that is not as useful as reporting the trends in hospitalizations over time. We have updated this paragraph to reflect hospitalizations and not the unique payers per month (which was used as the offset in our analysis).*

I don't think it is necessary for the authors to address this, but I thought it was interesting that in Figure S3, in each year from 2017 to 2019, there is a spike in hospitalizations in January. The counts fall throughout the first half of the year until they settle at about 50%-70% the January level. Much of this is driven by normal seasonal infectious disease patterns. Though, some of it might also be due to how OfficeAlly is constructed, or because of new insurance policies beginning in January. Line 186 states that hospitalizations from January to April 2020 dropped from 455,000 to 151,000. That sounds huge, but actually a lot of that is just the normal January to April drop seen in each of the previous three years. Of course, the analysis corrects for this and the rest of the results are presented as deviations from the expected counts. So, this isn't a problem. However, until the authors added Figure S3, I thought there was a far greater drop than

there truly was. It is

also interesting that in Figure S1, from 2019 to 2021, January 2020 has the most payers and April 2020 had the fewest. This also might have exaggerated the drop a bit over that four month period.

We are glad that the inclusion of Figure S3 has helped to contextualize this decline in hospitalizations at the start of each year.

Reviewer #2 (Remarks to the Author):

*Reviewer #2 signaled their newer reviewer comments with “**” and we have highlighted them in light grey to distinguish these remarks from previous reviewer response comments that were addressed.*

** Reviewer Responses: 1 for each general comment and then an additional response for each of the introduction, methods, results and discussion sections.

General comments:

This is an interesting epidemiological study of the estimated effects of the COVID-19 pandemic on non-COVID-19 hospitalization trends between the beginning of the pandemic and June 2021. The use of a large database with 3 years of hospitalization data prior to the start of the pandemic combined with a thorough analysis results in a convincing depiction of these trends.

We thank the reviewer for their positive review of our work.

** Reviewer Response: The revised article reads well and the authors have taken care to respond to the comments. I have a few further comments in response which can be found below.

***Thank you*

General Comment A) The most significant limitation is the lack of data beyond June 2021 – which means that there is no information on trends during the Delta and Omicron waves, which were both extremely significant, but also occurred in a population who had been living with the pandemic for over a year already (see attached image file). In addition, vaccination began in December 2020, and as a result most of this study’s period of interest is in the pre-vaccination period, which would obviously have a huge impact on population-level behavior.

Response A): Unfortunately, the last update to this database was provided in September 2021 and C19RDB informed us that there is a potential 3-month lag in reporting, we decided to err on the side of caution and end our analysis in June 2021. We realize that this excludes a large

proportion of the most recent epidemic waves but are limited by the database itself. We have a statement in the Methods section to indicate that study length is due to data limitations: “Reporting to this database was continually updated until September 2021...”

**** Reviewer Response:** I understood that the limitation was a data limitation – but this does not prevent it from being a limitation. Whilst this addition makes it clear that the study length is due to data limitations, it would be good to clarify in the introduction that the study aims to assess the relative changes in the incidence of hospitalizations by diagnostic category in the early-medium phase (first 18 months) of the COVID-19 pandemic. I think it may be prudent to also add in a sentence in the limitations section given that this clearly limits the scope of the interpretation of the results. In general, the framing of the clinical utility has been improved and as such this is less problematic in this version of the manuscript.

****Response:** *We have clarified this point in the introduction now, through the addition of the statement “In this study, we leverage a large national healthcare billing database to estimate relative changes in the incidence of hospitalizations by diagnostic category as the COVID-19 pandemic emerged and progressed through the early-medium phase of the pandemic (first 16 months) and the first waves of SARS-COV-2 variants. We have also added a sentence to the limitations section, as suggested: “Because the database was not updated after September 2021, analyses of the latest waves of the pandemic were not possible and later trends may vary from early trends in non-COVID-19 hospitalizations.”.*

General Comment B) The result of this limitation may be that this manuscript does not hold the practical utility the authors suggest, and the focus should be more on the interesting epidemiological findings of the detailed effects of a large pandemic on non-COVID hospital services. Furthermore, due to the methodology and types of data used, it is difficult to ascertain causality: a whole range of things changed with the onset of the pandemic and teasing apart which factor influence the drop is not possible from this type of analysis. As such, once again, the language could be focused more on describing the epidemiological findings than on possible causative mechanisms (specific points in the comments below).

Response B): We have tailored the manuscript according to the recommendations below so that the practical utility for other researchers and clinicians remains true. We feel this study offers useful insight for fellow researchers who are interested in key trends but can only access smaller datasets and thus cannot draw immediate conclusions on whether the trends they see may be meaningful for further exploration.

Language used in the Results and Discussion section is careful to describe association and not causation. We do not imply that any of these findings show causal mechanisms and take care to offer alternative explanations for certain findings in the Discussion section. We have further edited the manuscript to ensure causal language is avoided and secular trends in the data are described as needed. The discussion section attempts to contextualize the findings within the broader literature to better understand potential mechanisms at play; however, this does not imply causality.

Numerous studies have demonstrated that the effects of the COVID-19 pandemic on healthcare

Trends in non-COVID-19 hospitalizations; NCOMMS-22-16251A

utilization were most severe in the early phases of the pandemic. Our study of this critical period of the pandemic can be important to understand the short- and long-term consequences of these shifts in hospitalization trends.

**** Reviewer Response:** The results and discussion sections read better now, and contextualise well without implying causation. The language regarding the practical utility of the manuscript is now more appropriately focussed. A few minor detailed points are added below.

****Response:** Thank you!

Detailed comments:

Introduction:

Comment 1 Lines 72-75: The two articles referenced here were published at a much earlier stage in the pandemic. Do the results of the current manuscript (that show trends up until June 2021 – before Delta and Omicron and in a predominantly pre-vaccination period) really aid us in hospital preparation and resource allocation? If so, how?

Response 1 Lines 72-75: We see Reviewer 2's point that this analysis does not capture the most recent waves of the pandemic and have removed this sentence so as to not overstate our impact. We still feel this study is important because it covers the early pandemic period, prior to restructuring to accommodate capacity issues. This study is more specifically about disruptions during and after the unique crisis period at the start of the pandemic.

Comment 2 Lines 79-81: The analysis results are limited in their ability to inform such specific hypotheses – in order to inform the hypothesis for example that NPIs influenced hospitalizations numerous confounders would have to be dealt with, and this is out of the scope of the current article. Consider removing this sentence.

Response 2 Lines 79-81: We agree with Reviewer 2 on this point and have removed this sentence. Instead we focus on our ability to leverage multiple years of pre-pandemic data to aid in our estimations, the incorporations of public and private insurance patient data from the US and the ability to estimate changes in both common and uncommon causes of hospitalization, thereby being of interest to a range of specialties.

**** Reviewer Response:** Overall the introduction has improved. The only phrase that still jars is the phrasing “progressed through multiple waves of SARS-CoV2 variants”, which still in some way overstates the scope of the study. It would be more appropriate to phrase it “progressing through the first two major waves of SARS-CoV2 variants” or “and progressed through the first 18 months of the pandemic”.

****Response:** *We see this point and have adjusted the mentioned sentence to state “In this study, we leverage a large national healthcare billing database to estimate relative changes in the incidence of hospitalizations by diagnostic category as the COVID-19 pandemic emerged and*

progressed through the early-medium phase of the pandemic (first 16 months) and the first waves of SARS-COV-2 variants.”.

Methods:

Comment 3 Lines 114-118: Does this include primary as well as secondary diagnostic codes? The number of hospitalizations excluded (66,099) seems a little small on the basis that we are drawing from a population significantly larger than 13.5 million individuals who were hospitalized in this dataset. From the CDC website it appears as if about nearly 2.5 million hospitalizations occurred in the US by June 2021 (<https://www.cdc.gov/coronavirus/2019-ncov/covid-data/covidview/index.html>). Maybe some of the hospitalizations removed because they occurred within a 30-day window of prior hospitalization were COVID-19 related?

Response 3 Lines 114-118: We have reformatted the Supplemental Figure (now S2) that shows the data inclusion/exclusion process to clarify this point. Previously the COVID-19 cases were shown as being excluded later in the process and thus looked erroneously small. You are correct, some of the COVID hospitalizations were removed in the 30d-window of hospitalization.

Comment 4 Line 133: Please note that in Supplementary Table S2 it is written that subchapters with fewer than 500 diagnoses (rather than 5000) were excluded.

Response 4 Line 133: Thank you for pointing this out! We have fixed this mistake in the revised draft. Overall, the development of the Poisson models to predict the expected rates based on the previous three years seem to have been performed rigorously and provides a good estimated baseline for expected rates.

Response: We kindly thank the reviewer for this positive feedback

**** Reviewer Response:** The first half of the new flow chart is quite confusing. It might be clearer if those with missing diagnostic info are removed initially and then those with COVID-19 hospitalizations and then those with a hospitalization within 30d of previous. Other than that I have no further comments on the methods section.

****Response:** *We see what the author is referring to here. Technically, the removal of cases that are missing diagnostic info is happening prior to the removal of COVID-19 hospitalizations, in the initial data pull from OfficeAlly. We merely retained this descriptor from the prior version of this flowchart. We have removed this descriptor from the flowchart since it is assumed that all missing variables are removed in the initial data pull from OfficeAlly.*

Results:

Comment 5 Lines 171-177: The phrasing here is confusing. In general the monthly unique hospitalizations including the relevant trends should be described from Jan 2017 until Feb 2020 (including an additional mention of a large increase in Jan and Feb 2020 if indeed there was a big

Trends in non-COVID-19 hospitalizations; NCOMMS-22-16251A

jump to 455,000 from less than 400,000 in any month prior to 2020) and then a description of the general trends from March 2020 until June 2021. This information is not presented in a figure or table (at least no figure or table is referenced) and as such needs to be clearly stated here in full. It might be also useful to note here the number of excluded COVID-19 related hospitalizations during the pandemic period.

Response 5 Lines 171-177: We chose to report the difference in hospitalizations as such to highlight the secular changes in the database between Dec 2019 and Jan 2020. In this revision, we have added a supplemental figure (Supplemental Figure S3 inserted at the bottom of this document) to show the number of monthly all-cause hospitalizations (primary diagnostic code field only), and monthly COVID-19 hospitalizations (all diagnostic code fields).

Comment 6 Lines 181-183: It is confusing to report relative increases across all groups and then present an IRR of 0.96 which would represent a decrease. Consider specifically commenting on Conditions in the perinatal period as the only chapter that didn't show an increase.

Response 6 Lines 181-183: We have incorporated this feedback and rephrased those sentences as such: "All ICD-10 chapters experienced relative increases in incidence of hospitalizations during the pre-pandemic months of January and February 2020, with IRRs ranging from 1.08 to 1.21 in Jan 2020 (Figure 1; Supplemental Table S3), likely reflecting database reporting changes at the start of the new calendar year. The exception being conditions of the perinatal period, which had an estimated IRR of 0.96 (95%CI: 0.87, 1.06) in Jan 2020."

Comment 7 Lines 184-185: Likewise with March 2020, the perinatal period chapter is the same in March as February and as such this sentence is confusing.

Response 7 Lines 184-185: The relative decline reported in March 2020 is compared to previous years data (March 2019, March 2018, March 2017), not Feb 2020 data. Thus, conditions of the perinatal does still show a relative decline compared to March 2019. We did alter the statement to be "Beginning in March 2020, the relative incidence of hospitalization among all diagnostic chapters exhibited a moderate decrease compared to baseline." to clarify this point.

Comment 8 Lines 196-243: The description of the clusters is a bit clunky and could be streamlined. Maybe consider leading with the types of conditions in each cluster followed by the trends of that cluster of the pandemic. For example: Cluster A consists of the subchapters [list relevant subchapters]. These subchapters demonstrated a sharp relative decrease in hospitalizations early in the pandemic with rates below expected until [the end of 2020? June 2021?]. Then provide some specific details of interest such as rates of P&I. It may help with the flow and clarity of the clusters.

Response 8 Lines 196 – 243: Thank you for this advice. We have altered the Results section to introduce the subchapters first and then trends second as suggested.

Comment 9 Lines 218-225: Specifically, with cluster B, the phrasing is confusing – consider "the average IRR in this cluster remained at or below expected levels from March 2020 – May 2020 and exceeded expected levels for the remainder of the study period."

Trends in non-COVID-19 hospitalizations; NCOMMS-22-16251A

Response 9 Lines 218-225: We appreciate this feedback and have incorporated the suggested phrasing.

Comment 10 Lines 242-243: Please update to ‘between March 2020 and June 2021. Current phrasing ‘between March and June 2021’ is confusing

Response 10 Lines 242-243: We see how this is confusing and thank you for pointing this out. We have changed the sentence as suggested.

**** Reviewer Response:** Overall the results section reads well and is easier to follow. Regarding supplemental figure S3 I would recommend adding in a vertical line at January of each year so that the changes during the period of interest are clearly and it would be easier to gauge how much things changed year to year. Regarding comment 7 – my mistake.

****Response:** We have added a dashed line to aid in the interpretation of Supplementary Figure S3.

Discussion:

Comment 11 Lines 276-318: Generally, this section needs to be cut down. A significant proportion of the text is spent trying to explain what seem like possible artefacts in the data or things that the data cannot explain such as the trends in hospitalizations where the primary diagnosis is intellectual disabilities. It might be more effective to build a paragraph around ‘signals’ generated that could be leads for further research (is there a causal relationship between SARS-CoV-2 pandemic and hemolytic anemias/intellectual disabilities/adverse newborn outcomes). In addition, as a way of providing some validation to the results, it would be interesting to show that the number of childbirth coded diagnoses (i.e. sub-chapter) didn’t drop in the early pandemic (it should be unaffected overall as the authors mention) and that the drop in the pregnancy, childbirth and puerperium diagnoses occurred as a result of codes from the pregnancy and puerperium related codes.

Response 11 Lines 276-318: The discussion section has been edited to remove the paragraphs explaining relatively minor trends in the data (diagnostic codes for psychological disturbances) and to shift focus to the more impactful trends. Trends that we cannot readily explain (since the dataset doesn’t allow for interpretations of mechanisms), we have provided references for further reading but have limited our focus. Figure 2B shows that maternal delivery codes were not significantly altered early in the pandemic period, in contrast to the codes for newborn gestation disorder codes.

**** Reviewer Response:** Overall the discussion reads better, with an improved focus to the major trends and reduced focus on minor trends with unclear mechanisms that cannot really be teased apart with this dataset. The sensitivity analysis added is a good way of mitigating bias introduced by large payers and contributes to the robustness of the results presented.

****Response:** *Thank you. We were happy to include this sensitivity analysis and are pleased that the reviewers agree.*

Conclusion:

Comment 12 Lines 242-243 Lines 380-387: The conclusion here focusses a lot on the aspects that the analysis cannot actually answer – why the trends are what they are. An alternative focus for the conclusion could be that for the first time, by leveraging a large billing database in the US, a comprehensive high-level analysis of hospitalization by diagnosis has identified broad trends over the pandemic period. This is useful because...[insert reasons].

Response 12 Lines 242-243: We have incorporated this suggestion throughout the discussion section and switched the focus on aspects of why the analysis is important, rather than signals in the data that are challenging to explain.

Comment 13 Lines 387-388: It is unclear if the last sentence adds value to the manuscript and if so, how this will help minimize disruptions during future major health events. Recommend removing and leaving the reader to decide if the findings are of interest/use.

Response 13 Lines 387-388: We have removed the mentioned sentence as suggested.

**** Reviewer Response:** Overall the conclusion reads well and more accurately represents the findings of the study and its utility.

****Response:** *Thank you.*

Miscellaneous:

Supplementary figure S3 – The legend seems to be unedited: “shows the clusters of the subchapters and where I made the cut-offs – can update this to have clear demarcations of the cut-offs” – please correct.

Response: Apologies, we have fixed this mistake in this revision. Thank you for highlighting this.

Reviewer #3 (Remarks to the Author):

Thank you to the authors for the extent to which they have incorporated suggestions from my prior review into their manuscript.

The manuscript reads much better and is a clearer and more balanced article. There are a few

points remaining which I would encourage the authors to consider to enhance the clarity of the paper.

Further clarity for the reader as to what admissions with a primary diagnosis of ‘intellectual disabilities’, ‘physiological disturbances’ and ‘biomechanical lesions’ actually mean would also be very helpful. These are often quite broad or vague definitions, and the latter two are rarely used in my everyday practice at least! I still think it would be very helpful to mention the incidence of hospitalisation in each group to give the readers an idea of whether they constitute low or high volume conditions.

If anything, one of the most useful figures in the whole manuscript is figure S6 (in conjunction with Figure 1). This is the only place in the whole manuscript where I get an idea of the relative incidence of hospitalisations in each smaller diagnostic group. It shows me that many of the conditions referred to in the manuscript proper are rare (biomechanical lesions, intellectual disabilities, haemolytic anaemias, physiological disturbances) in comparison to the other diagnoses. Given these small numbers, the potential for statistical noise is high. There are higher volume conditions (such as URI and LRI, middle ear, mental disorders) that show profound trends that may be more robust to describe.

From the perspective of a policymaker, it is smaller fluctuations in more common conditions that are perhaps of most interest, rather than larger fluctuations in less common conditions. In the case of these rarer conditions, it is likely that this sort of ‘catch all’ administrative data study is not the best way to evaluate changes in such conditions and as such they should not occupy the volume of results and discussion they do.

Response: *To address this, we have removed sentences addressing the increase in intellectual disability disease/biomechanical lesion diagnoses from the Discussion Section. Now, this paragraph ends on the note that these strange elevations in IRR could be due to a number of (likely unrelated) mechanisms.*

****Minor points:****

I think it’s ‘genital tract’ rather than ‘genital track’.

Response: *This has been fixed.*

Looking at the colormaps, the centre point of the map (0.90-1.10) is yellow, and this alludes to the positive end of the scale, and perhaps suggests to the reader values greater than one. Perhaps have this centre value be white / grey so as to avoid confusion here.

Response: *We have decided to retain the light yellow color because white or gray might imply a lack of association and we want to maintain our color scale.*

Overall, the manuscript is clearer and more focussed, however I do have concerns about the emphasis placed on less common conditions where there may be greater uncertainty about the

Trends in non-COVID-19 hospitalizations; NCOMMS-22-16251A

estimates obtained and trends observed.

Response: *Thank you; we have tried to trim the Discussion section (as mentioned above) to place less emphasis on the trends in rarer conditions that we noted.*